# Denoising and Regularization via Exploiting the Structural Bias of Convolutional Generators

**Reinhard Heckel**
Dept. of Electrical and Computer Engineering
Technical University of Munich
`reinhard.heckel@tum.de`

**Mahdi Soltanolkotabi**
Dept. of Electrical and Computer Engineering
University of Southern California
`soltanol@usc.edu`

## Abstract

Convolutional Neural Networks (CNNs) have emerged as highly successful tools for image generation, recovery, and restoration. A major contributing factor to this success is that convolutional networks impose strong prior assumptions about natural images. A surprising experiment that highlights this architectural bias towards natural images is that one can remove noise and corruptions from a natural image without using any training data, by simply fitting (via gradient descent) a randomly initialized, over-parameterized convolutional generator to the corrupted image. While this over-parameterized network can fit the corrupted image perfectly, surprisingly after a few iterations of gradient descent it generates an almost uncorrupted image. This intriguing phenomena enables state-of-the-art CNN-based denoising and regularization of other inverse problems. In this paper we attribute this effect to a particular architectural choice of convolutional networks, namely convolutions with fixed interpolating filters. We then formally characterize the dynamics of fitting a two layer convolutional generator to a noisy signal and prove that early-stopped gradient descent denoises/regularizes. Our proof relies on showing that convolutional generators fit the structured part of an image significantly faster than the corrupted portion.

## 1 Introduction

Convolutional neural networks are extremely popular for image generation. The majority of image generating networks is convolutional, ranging from deep convolutional Generative Adversarial Networks (DC-GANs) (Radford et al., 2015) to the U-Net (Ronneberger et al., 2015). It is well known that convolutional neural networks incorporate implicit assumptions about the signals they generate, such as pixels that are close being related. This makes them particularly well suited for modeling distributions of images. It is less well known, however, that those prior assumptions build into the architecture are so strong that convolutional neural networks are useful even without ever being exposed to training data.

The latter was first shown in the Deep Image Prior (DIP) paper (Ulyanov et al., 2018). Ulyanov et al. (2018) observed that when 'training' a standard convolutional auto-encoder such as the popular U-net (Ronneberger et al., 2015) on a single noisy image and regularizing by early stopping, the network denoises the image with excellent performance. This is based on the empirical observation that un-trained convolutional auto-decoders fit a single natural image faster when optimized with gradient descent than pure noise. Many elements of an auto-encoder, however are irrelevant for this effect: A more recent paper (Heckel & Hand, 2019) proposed a much simpler image generating network, termed the deep decoder. This network can be seen as the relevant part of a convolutional generator architecture to function as an image prior, and can be obtained from a standard convolutional autoencoder by removing the encoder, the skip connections, and perhaps most notably, the trainable convolutional filters of spatial extent larger than one. Thus, the deep decoder does not use learned or trainable convolutional filters like conventional convolutional networks do, and instead only uses convolutions with fixed convolutional kernels to generate an image.

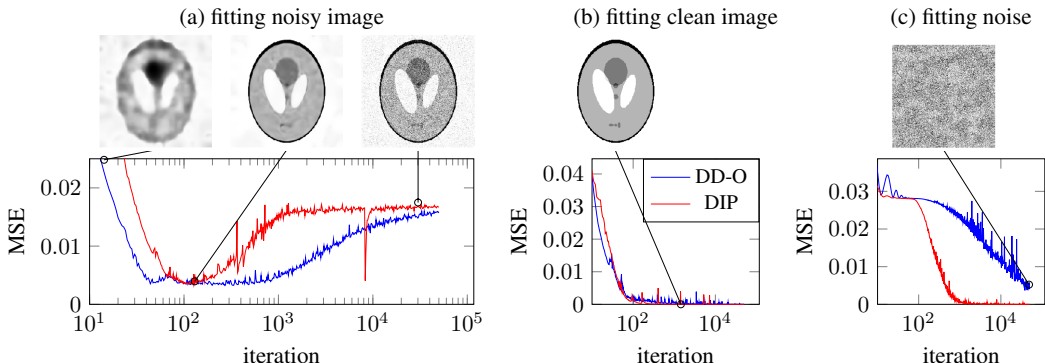

Figure 1: Fitting an over-parameterized Deep Decoder (DD-O) and the deep image prior (DIP) to a (a) noisy image, (b) clean image, and (c) pure noise. Here, MSE denotes Mean Square Error of the network output with respect to the clean image in (a) and fitted images in (b) and (c). While the network can fit the noise due to over-parameterization, it fits natural images in significantly fewer iterations than noise. Hence, when fitting a noisy image, the image component is fitted faster than the noise component which enables denoising via early stopping.

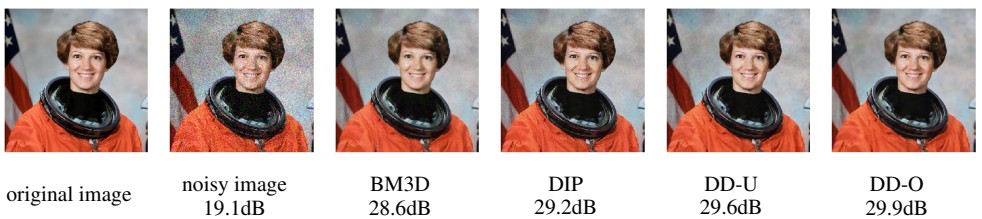

Figure 2: Denoising with BM3D and various convolutional generators. The relative ranking of algorithms in this picture is representative and maintained on a larger set of test images (see Appendix A). DD-U is an under-parameterized deep decoder. DIP and DD-O are over-parameterized convolutional generators and with early stopping outperform the BM3D algorithm, the next best method that does not require training data.

In this paper, we study such a simple, untrained convolutional network theoretically. We consider the over-parameterized regime where the network has sufficiently many parameters to represent an arbitrary image (including noise) perfectly and show that:

*Fitting convolutional generators via early stopped gradient descent provably denoises "natural" images.*

To prove this statement, we characterize how the network architecture governs the dynamics of fitting over-parameterzed networks to a single (noisy) image. In particular we prove:

*Convolutional generators optimized with gradient descent fit natural images faster than noise.*

We depict this phenomena in Figure 1 where we fit a randomly initialized over-parameterized convolutional generator to an image via running gradient descent on the objective $\mathcal{L}(\mathbf{C}) = \|G(\mathbf{C}) - \mathbf{y}\|_2^2$. Here, $G(\mathbf{C})$ is the convolutional generator with weight parameters $\mathbf{C}$, and $\mathbf{y}$ is either a noisy image, a clean image, or noise. This experiment demonstrates that an over-parameterized convolutional network fits a natural image (Figure 1b) much faster than noise (Figure 1c). Thus, when fitting the noisy image (Figure 1a), early stopping the optimization enables image denoising. This effect is so strong that it gives state-of-the-art denoising performance (see Figure 2 and Appendix A). Beyond denoising, this effect also enables significant improvements in regularizing a variety of inverse problems such as compressive sensing (Veen et al., 2018; Heckel, 2019).

## 1.1 CONTRIBUTIONS AND OVERVIEW OF RESULTS

In this paper we take a step towards understanding why fitting a convolutional generator via early stopped gradient descent leads to such surprising denoising and regularization capabilities.

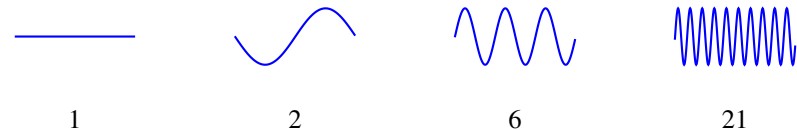

|  1  |  2  |  6  |  21  |

Figure 3: The 1st, 2nt, 6th, and 21th trigonometric basis functions in dimension $n = 300$.

- We first show experimentally that this denoising phenomena is primarily attributed to convolutions with fixed interpolation kernels, typically implicitly implemented by bi-linear upsampling operations in the convolutional network.

- We then show that fitting an over-parameterized convolutional networks (with fixed convolutional filters) via early stopped gradient descent to a signal provably denoises it.

  Specifically, let $\mathbf{x} \in \mathbb{R}^n$ be a smooth signal that can be represented exactly as a linear combination of the $p$ orthogonal trigonometric functions of lowest frequency (defined in equation (5), see Figure 3 for a depiction). A smooth signal is a good model for a natural image since natural images are well approximated by low-frequency components. Specifically, Figure 4 in (Simoncelli & Olshausen, 2001) shows that the power spectrum of a natural image (i.e., the energy distribution by frequency) decays rapidly from low frequencies to high frequencies. In contrast, Gaussian noise has a flat power spectrum.

  Our goal is to obtain an estimate of this signal from a noisy observation $\mathbf{y} = \mathbf{x} + \mathbf{z}$ where $\mathbf{z} \sim \mathcal{N}(0, \frac{\varsigma^2}{n}\mathbf{I})$ is Gaussian noise. Let $\hat{\mathbf{y}} = G(\mathbf{C}_\tau)$ be the estimate obtained from early stopping the fitting of a two-layer convolutional generator to $\mathbf{y}$. We prove this estimate achieves the denoising rate

  $$\|\hat{\mathbf{y}} - \mathbf{x}\|_2^2 \le c\frac{p}{n}\varsigma^2,$$

  with $c$ a constant. This rate is optimal up to a constant factor.

- Our denoising result follows from a detailed analysis of gradient descent applied to fitting a convolutional generator $G(\mathbf{C})$ to a noisy signal $\mathbf{y} = \mathbf{x} + \mathbf{z}$ with $\mathbf{x}$ representing the signal and $\mathbf{z}$ the noise. We show that there are weights $\sigma_1, \ldots, \sigma_n$ associated with the trigonometric basis function that only depend on the convolutional filter used, and which typically decay quickly from the low-frequency to the high-frequency trigonometric basis functions. These weights in turn determine the speed at which the associated components of the signal are fitted. Specifically, we show that the dynamics of gradient descent are approximately given by

  $$G(\mathbf{C}_\tau) - \mathbf{x} \approx \underbrace{\sum_{i=1}^{n} \mathbf{w}_i \langle \mathbf{w}_i, \mathbf{x} \rangle (1 - \eta\sigma_i^2)^\tau}_{\text{error in fitting signal}} + \underbrace{\sum_{i=1}^{n} \mathbf{w}_i \langle \mathbf{w}_i, \mathbf{z} \rangle ((1 - \eta\sigma_i^2)^\tau - 1)}_{\text{fit of noise}}.$$

  The convolutional filters commonly used are such that the weights $\sigma_1, \ldots, \sigma_n$ decays quickly, implying that low-frequency components in the trigonometric expansion are fitted significantly faster than high frequency ones. So if the signal mostly consists of low-frequency components, we can choose an early stopping time such that the error in fitting the signal is very low, and thus the signal part is well described, whereas at the same time only a small part of the noise, specifically the part aligned with the low-frequency components has been fitted.

## 2 CONVOLUTIONAL GENERATORS

A convolutional generator maps an input tensor $\mathbf{B}_0$ to an image only using upsampling and convolutional operations, followed by channel normalization (a special case of batch normalization) and applications of non-linearities, see Figure 4. All previously mentioned convolutional generator networks (Radford et al., 2015; Ronneberger et al., 2015) including the networks considered in the DIP paper (Ulyanov et al., 2018) primarily consist of those operations.

For motivating the architecture of the convolutional generators studied in this paper, we first demonstrate in Section 2.1 that convolutions with fixed interpolation filters are critical to the denoising

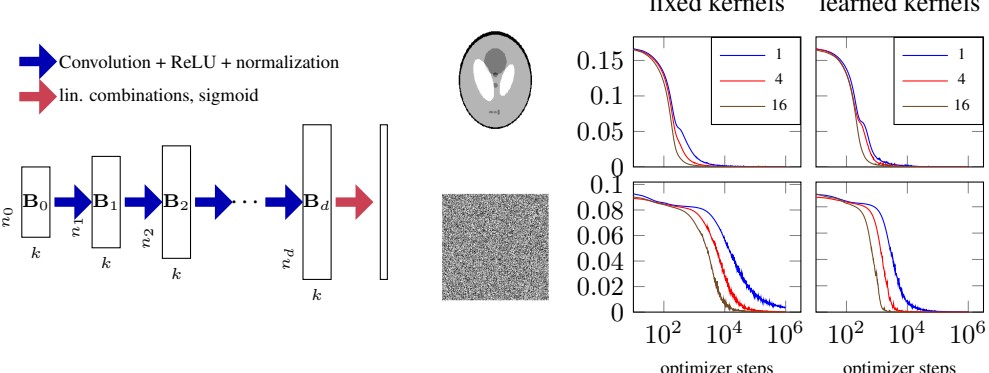

Figure 4: **Left panel:** Convolutional generators. The output is generated through repeated convolutional layers, channel normalization, and applying ReLU non-linearities. **Right panel:** Fitting the phantom MRI and noise with different architectures of depth $d = 5$, for different number of over-parameterization factors (1,4, and 16). Gradient descent on convolutional generators involving fixed convolutional matrixes fit an image significantly faster than noise.

performance with early stopping. Specifically, we empirically show that convolutions with *fixed* convolutional kernels are critical for convolutional generators to fit natural images faster than noise. In Section 2.2 we formally introduce the class of convolutional generators studied in this paper, and finally, in Section 2.3, we introduce a minimal convolutional architecture which is the focus of our theoretical results.

## 2.1 THE IMPORTANCE OF FIXED CONVOLUTIONAL FILTERS

Convolutions with fixed convolutional kernels are critical for denoising with early stopping, because they are critical for the phenomena that natural images are fitted significantly faster than noise. To see this, consider the experiment in Figure 4 in which we fit an image and noise by minimizing the least-squares loss via gradient descent with i) a convolutional generator with only fixed convolutional filters (see Section 2.2 below for a precise description) and ii) a conventional convolutional generator with trainable convolutional filters (essentially the architecture from the popular DC-GAN generators, see Appendix B for details and additional numerical evidence). As illustrated in Figure 4, the convolutional network with *fixed filters* fits the natural image much faster than noise, whereas the network with learned convolutional filters, only fits it slightly faster, and this effect vanishes as the network becomes highly overparameterized. Thus, fixed convolutional filters enable un-trained convolutional networks to function as highly effective image priors. We note that the upsampling operation present in most architectures implicitly incorporates a convolution with a *fixed* convolutional (interpolation) filter.

## 2.2 ARCHITECTURE OF CONVOLUTIONAL GENERATOR WITH FIXED CONVOLUTIONS

In this section, we describe the architecture of the deep decoder (Heckel & Hand, 2019), a convolutional network with *fixed* convolutional operators only. In this architecture, the channels in the $(i+1)$-th layer are given by

$$\mathbf{B}_{i+1} = \mathrm{cn}(\mathrm{ReLU}(\mathbf{U}_i\mathbf{B}_i\mathbf{C}_i)), \quad i = 0, \ldots, d-1,$$

and finally, the output of the $d$-layer network is formed as

$$\mathbf{x} = \mathbf{B}_d\mathbf{C}_{d+1}.$$

Here, the coefficient matrices $\mathbf{C}_i \in \mathbb{R}^{k \times k}$ and $\mathbf{C}_{d+1} \in \mathbb{R}^{k \times k_{\mathrm{out}}}$ contain the weights of the network. The number of channels, $k$, determines the number of weight parameters of the network, given by $dk^2 + k_{\mathrm{out}}k$. Each column of the tensor $\mathbf{B}_i\mathbf{C}_i \in \mathbb{R}^{n_i \times k}$ is formed by taking linear combinations of the channels of the tensor $\mathbf{B}_i$ in a way that is consistent across all pixels, and the ReLU activation function is given by $\mathrm{ReLU}(t) = \max(0, t)$. Then, $\mathrm{cn}(\cdot)$ performs the channel normalization operation which normalizes each channel individually and can be viewed as a special case of the popular batch normalization operation (Ioffe & Szegedy, 2015).

The operator $\mathbf{U}_i \in \mathbb{R}^{n_{i+1} \times n_i}$ is a tensor implementing an upsampling and most importantly a convolution operation with a fixed kernel. This fixed kernel was chosen in all experiments above as a triangular kernel so that $\mathbf{U}$ performs bi-linear 2x upsampling (this is the standard implementation in the popular software packages pytorch and tensorflow). As mentioned earlier this convolution with a *fixed kernel* is critical for fitting natural images faster than complex ones.

## 2.3 TWO LAYER CONVOLUTIONAL GENERATOR STUDIED THEORETICALLY IN THIS PAPER

The simplest model to study the denoising capability of convolutional generators and the phenomena that a natural image is fitted faster than a complex one theoretically is a network with only one hidden layers and one output channel i.e., $G(\mathbf{C}) \in \mathbb{R}^n$. Then, the generator becomes

$$G(\mathbf{C}) = \text{ReLU}(\mathbf{U}\mathbf{B}_0\mathbf{C}_0)\mathbf{c}_1,$$

where $\mathbf{U} \in \mathbb{R}^{n \times n}$ is a circulant matrix that implements a convolution with a filter $\mathbf{u}$. In this paper we consider the over-parameterized regime where $k \geq 2n$. Note that scaling the $i$-th column of $\mathbf{C}_0$ with a non-negativ factor is equivalent to scaling the $i$-th entry of the output weights $\mathbf{c}_1$ with the same factor. We therefore fix the output weights to $\mathbf{c}_1 = \mathbf{v}$, where $\mathbf{v} = [1, \ldots, 1, -1, \ldots, -1]/\sqrt{k}$. Next, note that both the generators with the fixed output weights and non-fixed output weights have the same range in the regime $k \geq 2n$. Next, because the matrix $\mathbf{B}_0$ is Gaussian, with probability one, it has full rank and thus spans $\mathbb{R}^n$. It follows that optimizing over the parameter $\mathbf{C}_0$ in $\mathbf{C} = \mathbf{B}_0\mathbf{C}_0$ is equivalent to optimizing over the matrix $\mathbf{C} \in \mathbb{R}^{n \times k}$. We therefore consider the generator

$$G(\mathbf{C}) = \text{ReLU}(\mathbf{U}\mathbf{C})\mathbf{v}, \tag{1}$$

where $\mathbf{v} = [1, \ldots, 1, -1, \ldots, -1]/\sqrt{k}$ is fixed and $\mathbf{C} \in \mathbb{R}^{n \times k}$ is the new coefficient matrix we optimize over. Figure 11 in the appendix shows that even this simple two-layer convolutional network fits a simple image faster than noise. This is the simplest model in which the phenomena that a convolutional networks fits structure faster than noise can reliably be observed. As a consequence, the dynamics of training the model (1) are the focus of the remainder of this paper.

## 3 WARMUP: DYNAMICS OF GRADIENT DESCENT ON LEAST SQUARES

As a prelude for studying the dynamics of fitting convolutional generators via a *non-linear* least square problem, we study the dynamics of gradient descent applied to a *linear* least squares problem. We demonstrate how early stopping can lead to denoising capabilities even with a simple linear model. We consider a least-squares problem of the form

$$\mathcal{L}(\mathbf{c}) = \frac{1}{2}\|\mathbf{y} - \mathbf{J}\mathbf{c}\|_2^2,$$

and study gradient descent with a constant step size $\eta$ starting at $\mathbf{c}_0 = \mathbf{0}$. The updates are given by

$$\mathbf{c}_{\tau+1} = \mathbf{c}_\tau - \eta\nabla\mathcal{L}(\mathbf{c}_\tau), \quad \nabla\mathcal{L}(\mathbf{c}) = \mathbf{J}^T(\mathbf{J}\mathbf{c} - \mathbf{y}).$$

The following simple proposition characterizes the trajectory of gradient descent.

**Proposition 1.** *Let $\mathbf{J} \in \mathbb{R}^{n \times m}$ be a matrix with left singular vectors $\mathbf{w}_1, \ldots, \mathbf{w}_n \in \mathbb{R}^n$ and corresponding singular values $\sigma_1 \geq \sigma_2 \geq \ldots \geq \sigma_n$. Then the residual after $\tau$ steps, $\mathbf{r}_\tau = \mathbf{y} - \mathbf{J}\mathbf{c}_\tau$, of gradient descent starting at $\mathbf{c}_0 = 0$ is*

$$\mathbf{r}_\tau = \sum_{i=1}^n \mathbf{w}_i \langle \mathbf{w}_i, \mathbf{y} \rangle (1 - \eta\sigma_i^2)^\tau.$$

Suppose that the signal $\mathbf{y}$ lies in the column span of $\mathbf{J}$, and that the stepsize is chosen sufficiently small (i.e., $\eta \leq 1/\|\mathbf{J}\|^2$). Then, by Proposition 1, gradient descent converges to a zero-loss solution and thus fits the signal perfectly. More importantly, gradient descent fits the components of $\mathbf{y}$ corresponding to large singular values faster than it fits the components corresponding to small singular values.

To explicitly show how this observation enables regularization via early stopped gradient descent, suppose our goal is to find a good estimate of a signal $\mathbf{x}$ from a noisy observation

$$\mathbf{y} = \mathbf{x} + \mathbf{z},$$

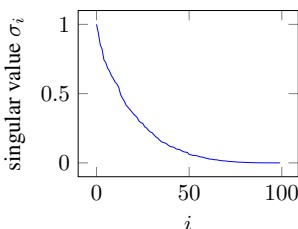 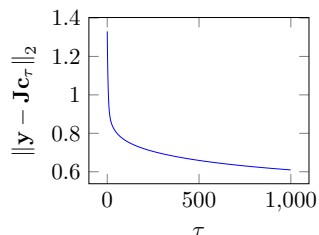 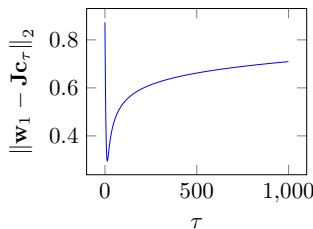

Figure 5: Gradient descent on the least squares problem of minimizing $\|\mathbf{y} - \mathbf{Jc}_\tau\|^2$, where $\mathbf{J} \in \mathbb{R}^{100 \times 100}$ has decaying singular values (left panel) and the observation is the sum of a signal component, equal to the leading singular vector $\mathbf{w}_1$ of $\mathbf{J}$, and a noisy component $\mathbf{z} \sim \mathcal{N}(0, (1/n)\mathbf{I})$, i.e., $\mathbf{y} = \mathbf{w}_1 + \mathbf{z}$. The signal component $\mathbf{w}_1$ is fitted significantly faster than the other components (right panel), thus early stopping enables denoising.

where the signal $\mathbf{x}$ lies in a signal subspace that is spanned by the $p$ leading left-singular vectors of $\mathbf{J}$. Then, by Proposition 1, the signal estimate after $\tau$ iterations, $\mathbf{Jc}_\tau$, obeys

$$\|\mathbf{Jc}_\tau - \mathbf{x}\|_2 \le (1 - \eta\sigma_p^2)^\tau \|\mathbf{x}\|_2 + E(\mathbf{z}), \quad E(\mathbf{z}) := \sqrt{\sum_{i=1}^n ((1 - \eta\sigma_i^2)^\tau - 1)^2 \langle \mathbf{w}_i, \mathbf{z} \rangle^2}. \quad (2)$$

Thus, after a few iterations most of the signal has been fitted (i.e., $(1 - \eta\sigma_p)^\tau$ is small). Furthermore, if we assume that the ratio $\sigma_p/\sigma_{p+1}$ is sufficiently large so that the spread between the two singular values separating the signal subspace from the rest is sufficiently large, most of the noise outside the signal subspace has not been fitted (i.e., $((1 - \eta\sigma_i^2)^\tau - 1)^2 \approx 0$ for $i = p + 1, \dots, n$).

In particular, suppose the noise vector has a Gaussian distribution given by $\mathbf{z} \sim \mathcal{N}(0, \frac{\varsigma^2}{n}\mathbf{I})$. Then $E(\mathbf{z}) \approx \varsigma\sqrt{\frac{p}{n}}$ so that after order $\tau = \log(\epsilon)/\log(1 - \eta\sigma_p^2)$ iterations, with high probability,

$$\|\mathbf{Jc}_\tau - \mathbf{x}\|_2 \le \epsilon \|\mathbf{x}\|_2 + c\varsigma\sqrt{\frac{p}{n}}.$$

This demonstrates, that provided the signal lies in a subspace spanned by the leading singular vectors, early stoped gradient descent reaches the optimal denoising rate of $\varsigma\sqrt{p/n}$ after a few iterations. See Figure 5 for a numerical example demonstrating this phenomena.

## 4 DYNAMICS OF GRADIENT DESCENT ON CONVOLUTIONAL GENERATORS

We are now ready to study the implicit bias of gradient descent towards natural/structured images theoretically. Consider the two-layer network (introduced in Section 2.3) of the form

$$G(\mathbf{C}) = \text{ReLU}(\mathbf{UC})\mathbf{v},$$

where $\mathbf{v} = [1, \dots, 1, -1, \dots, -1]/\sqrt{k}$, and with weight parameter $\mathbf{C} \in \mathbb{R}^{n \times k}$, and recall that $\mathbf{U}$ is a circulant matrix implementing a convolution with a kernel $\mathbf{u} \in \mathbb{R}^n$. We fit the generator to a signal $\mathbf{y} \in \mathbb{R}^n$ by minimizing the non-linear least squares objective

$$\mathcal{L}(\mathbf{C}) = \frac{1}{2}\|\mathbf{y} - G(\mathbf{C})\|_2^2 \quad (3)$$

with (early-stopped gradient) descent with a constant stepsize $\eta$ starting at a random initialization $\mathbf{C}_0$ of the weights. The iterates are given by

$$\mathbf{C}_{\tau+1} = \mathbf{C}_\tau - \eta\nabla\mathcal{L}(\mathbf{C}_\tau). \quad (4)$$

In our warmup section on linear least squares we saw that the singular vectors and values of the matrix $\mathbf{J}$ determine the speed at which different components of the noisy signal $\mathbf{y}$ are fitted by gradient descent. The main insight that enables us to extend this intuition to the nonlinear case is that the role of the matrix $\mathbf{J}$ can be replaced with the Jacobian of the generator, defined as $\mathcal{J}(\mathbf{C}) := \frac{\partial}{\partial \mathbf{C}}G(\mathbf{C})$.

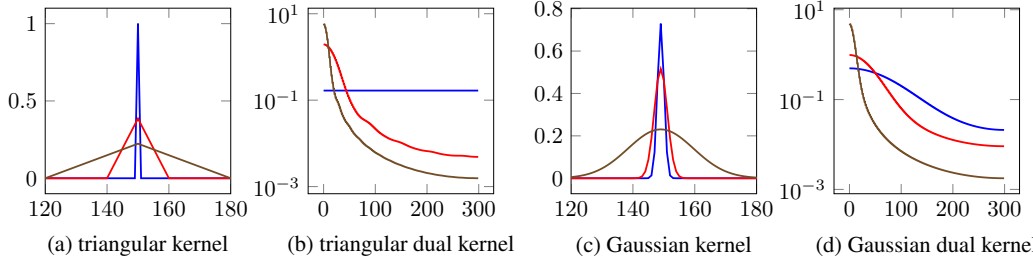

Figure 6: Triangular and Gaussian kernels and the weights associated to low-frequency trigonometric functions they induce, for a generator network of output dimension $n = 300$. The wider the kernels are, the more the weights are concentrated towards the low-frequency components of the signal.

Contrary to the linear least squares problem, however, in the nonlinear case, the Jacobian is not constant and changes across iterations. Nevertheless, we show that the eigen-values and vectors of the Jacobian at the random initialization govern the dynamics of fitting the network throughout the iterative updates.

For the two-layer convolutional generator that we consider, the left eigenvectors of the Jacobian mapping can be well approximated by the trigonometric basis functions, defined below, throughout the updates. Interestingly, the form of these eigenvectors only depends on the network architecture and not the convolutional kernel used.

**Definition 1.** *The trigonometric basis functions* $\mathbf{w}_1, \ldots, \mathbf{w}_n$ *are defined as*

$$[\mathbf{w}_{i+1}]_j = \frac{1}{\sqrt{n}} \begin{cases} 1 & i = 0 \\ \sqrt{2} \cos(2\pi j i / n) & i = 1, \ldots, n/2 - 1 \\ (-1)^j & i = n/2 \\ \sqrt{2} \sin(2\pi j i / n) & i = n/2 + 1, \ldots, n - 1 \end{cases}. \tag{5}$$

*Figure 3 depicts some of these eigenvectors.*

In addition to the left eigenvectors we can also approximate the spectrum of the Jacobian throughout the updates by an associated filter/kernel that only depends on the original filter/kernel used in the network architecture.

**Definition 2** (Dual kernel). *Associated with a kernel* $\mathbf{u} \in \mathbb{R}^n$ *we define the dual kernel* $\boldsymbol{\sigma} \in \mathbb{R}^n$ *as*

$$\boldsymbol{\sigma} = \|\mathbf{u}\|_2 \sqrt{\left| \mathbf{F} g \left( \frac{\mathbf{u} \circledast \mathbf{u}}{\|\mathbf{u}\|_2^2} \right) \right|} \quad \text{with} \quad g(t) = \frac{1}{2} \left( 1 - \frac{\cos^{-1}(t)}{\pi} \right) t.$$

*Here, for two vectors* $\mathbf{u}, \mathbf{v} \in \mathbb{R}^n$, $\mathbf{u} \circledast \mathbf{v}$ *denotes their circular convolution, the scalar non-linearity $g$ is applied entrywise, and $\mathbf{F}$ is the discrete Fourier transform matrix.*

In Figure 6, we depict two commonly used interpolation kernels $\mathbf{u}$, namely a triangular and a Gaussian kernel (recall that the standard upsampling operator is a convolution with a triangle), along with the induced dual kernel $\boldsymbol{\sigma}$. The figure shows that the dual kernel $\boldsymbol{\sigma}$ induced by these kernels has a few large values associated with the low frequency trigonometric functions, and the other values associated with high frequencies are very small.

With these definitions in place we are ready to state our main denoising result. A denoising result requires a signal model—we assume a low-frequency signal $\mathbf{x}$ that can be represented as a linear combination of the first $p$-trigonometric basis functions. Note that this is a good model for a natural image since natural images are well approximated by low-frequency components. Specifically, Figure 4 in (Simoncelli & Olshausen, 2001) shows that the power spectrum of a natural image (i.e., the energy distribution by frequency) decays rapidly from low frequencies to high frequencies. In contrast, Gaussian noise has a flat power spectrum.

**Theorem 1** (Denoising with early stopping). *Let* $\mathbf{x} \in \mathbb{R}^n$ *be a signal in the span of the first* $p$ *trigonometric basis functions, and consider a noisy observation*

$$\mathbf{y} = \mathbf{x} + \mathbf{z},$$

*where* $\mathbf{z}$ *is Gaussian noise with distribution* $\mathcal{N}(\mathbf{0}, \frac{\varsigma^2}{n}\mathbf{I})$, *for some variance* $\varsigma^2 \geq 0$. *To denoise this signal, we fit a two layer generator network* $G(\mathbf{C}) = \mathrm{ReLU}(\mathbf{UC})\mathbf{v}$ *with,*

$$k \geq C_{\mathbf{u}}n/\epsilon^8, \tag{6}$$

*channels, for some* $\epsilon > 0$, *and with convolutional kernel* $\mathbf{u}$ *of the convolutional operator* $\mathbf{U}$ *and associated dual kernel* $\boldsymbol{\sigma}$, *to the noisy signal* $\mathbf{y}$. *Here,* $C_{\mathbf{u}}$ *a constant only depending on the convolutional kernel* $\mathbf{u}$. *Then, with probability at least* $1 - e^{-k^2} - \frac{1}{n^2}$, *the reconstruction error obtained after* $\tau = \log(1 - \sqrt{p/n})/\log(1 - \eta\sigma_{p+1}^2)$ *iterations of gradient descent (4) with step size* $\eta \leq \frac{1}{\|\mathbf{Fu}\|_\infty^2}$ *($\mathbf{Fu}$ is the Fourier transform of $\mathbf{u}$) starting from* $\mathbf{C}_0$ *with i.i.d.* $\mathcal{N}(0, \omega^2)$, *entries,* $\omega \propto \frac{\|\mathbf{y}\|_2}{\sqrt{n}}$, *is bounded by*

$$\|G(\mathbf{C}_\tau) - \mathbf{x}\|_2 \leq (1 - \eta\sigma_p^2)^\tau \|\mathbf{x}\|_2 + \varsigma\sqrt{\frac{2p}{n}} + \epsilon\|\mathbf{y}\|_2. \tag{7}$$

Note that for this choice of stopping time, provided that dual kernel decays sharply around the $p$-th singular value, the first term in the bound (7) (i.e., $(1 - \eta\sigma_p^2)^\tau \approx 0$) essentially vanishes and the error bound becomes $O(\varsigma\sqrt{\frac{p}{n}})$. The dual kernel decays sharply around the leading eigenvalues provided the kernel is for example a sufficiently wide triangular or Gaussian kernel (see Figure 6).

This result demonstrates that when the noiseless signal $\mathbf{x}$ is sufficiently structured (e.g. contains only the $p$ lowest frequency components in the trigonometric basis) and the convolutional generator has sufficiently many channels, then early stopped gradient descent achieves a near optimal denoising performance proportional to $\varsigma\sqrt{\frac{p}{n}}$. This theorem is obtained from a more general result stated in the appendix which characterizes the evolution of the reconstruction error obtained by the convolutional generator.

**Theorem 2** (Reconstruction dynamics of convolutional generators). *Consider the setting and assumptions of Theorem 1 but now with a fixed noise vector* $\mathbf{z}$, *and without an explicit stopping time. Then, for all iterates* $\tau$ *obeying* $\tau \leq \frac{100}{\eta\sigma_p^2}$ *and provided that* $k \geq C_{\mathbf{u}}n/\epsilon^8$, *for some* $\epsilon \in (0, \frac{\sigma_p}{\sigma_1}]$, *with probability at least* $1 - e^{-k^2} - \frac{1}{n^2}$, *the reconstruction error obeys*

$$\|G(\mathbf{C}_\tau) - \mathbf{x}\|_2 \leq (1 - \eta\sigma_p^2)^\tau \|\mathbf{x}\|_2 + \sqrt{\sum_{i=1}^n ((1 - \eta\sigma_i^2)^\tau - 1)^2 \langle \mathbf{w}_i, \mathbf{z} \rangle^2} + \epsilon\|\mathbf{y}\|_2.$$

This theorem characterizes the reconstruction dynamics of convolutional generators throughout the updates. In particular, it explains why convolutional generators fit a natural signal significantly faster than noise, and thus early stopping enables denoising and regularization. To see this, note that as mentioned previously each of the basis functions $\mathbf{w}_i$ has a (positive) weight $\sigma_i > 0$ associated with it that only depends on the convolutional kernel used in the architecture (through the definition of the dual kernel). These weights determine how fast the different components of the noisy signal are fitted by gradient descent. As we demonstrated earlier in Figure 6, for typical convolutional filters those weights decay very quickly from low to high frequency basis functions. As a result, when the signal $\mathbf{x}$ is sufficiently structured (i.e. lies in the range of the $p$ trigonometric functions with lowest frequencies), after a few iterations most of the signal is fitted (i.e., $(1 - \eta\sigma_p^2)^\tau$ is small), while most of the noise has not been fitted (i.e., $((1 - \eta\sigma_i^2)^\tau - 1)^2 \approx 0$ for $i = p + 1, \ldots, n$). Thus, early stopping achieves denoising.

The proof, provided in the appendix, is based on associating the following linear least-squares problem with the non-linear least squares problem (3):

$$\mathcal{L}_{\mathrm{lin}}(\mathbf{c}) = \frac{1}{2}\|G(\mathbf{C}_0) + \mathbf{J}(\mathbf{c} - \mathrm{vect}(\mathbf{C}_0)) - \mathbf{y}\|_2^2.$$

Here, $\text{vect}(\mathbf{C}_0) \in \mathbb{R}^{kn}$ is a vectorized version of the matrix $\mathbf{C}_0$, and $\mathbf{J} \in \mathbb{R}^{n \times nk}$ approximates the Jacobian of $G(\mathbf{C})$ around the initialization $\mathbf{C}_0$. The linear least-squares problem is obtained by linearizing the non-linear problem around the initialization. In the proof we show that if i) the network is sufficiently over-parameterized and if ii) the network is randomly initialized, then the linearized problem is a good approximation of the non-linear problem. We then show that the singular values of the Jacobian $\mathbf{J}$ are the trigonometric basis functions, and the singular values are the values of the dual kernel. The proof is then concluded by characterizing the trajectory of gradient descent applied to the linear least-squares problem above.

## 4.1 MULTILAYER NETWORKS AND MODERATE OVER-PARAMETERIZATION

Our theoretical results rely on the finding that for over-parameterized single hidden-layer networks, the leading singular vectors of the Jacobian are the trigonometric functions throughout all iterations, and that the associated weights (i.e., the singular values) are concentrated towards the low frequency components. In this regime, the dynamics are well approximated by a linear model. This general strategy can be extended to multi-layer networks, however if the network is not highly over-parameterized, an associated linear model might not be a good approximation.

In order to understand whether our finding of low-frequency components being being fitted faster than high-frequency ones carries over to multi-layer networks and to the moderately over-parameterized regime, in this section, we study muli-layer networks in the moderately overparameterized regime numerically. We show that for a multilayer network, the spectrum of the Jacobian is concentrated towards singular vectors/functions that are similar to the low-frequency components. We also show that throughout training those functions do vary, albeit the low frequency components do not change significantly and the spectrum remains concentrated towards the low frequency components. This shows that the implications of our theory continue to apply to muli-layer networks.

In more detail, we take a standard one dimensional deep decoder with $d = 4$ layers with output in $\mathbb{R}^{512}$ and with $k = 64$ channels in each layer. Recall that the standard one dimensional decoder obtains layer $i+1$ from layer $i$ by linearly combining the channels of layer $i$ with learnable coefficients followed by linear upsampling (which involves convolution with the triangular kernel $[1/2, 1, 1/2]$). The number of parameters is $d \times k^2 = 32 \cdot 512$, so the network is over-parameterized by a factor of 32. In Figure 7, we display the singular values as well as the leading singular vectors/function of the Jacobian at initialization ($t = 1$) and after $t = 50$ and $t = 3500$ iterations of gradient descent. As can be seen the leading singular vectors ($s = 1-5$) are close to the trigonometric basis functions and do not change dramatically throughout training. The singular vectors corresponding to increasingly smaller singular values ($s = 20, 50, 100, 150$) contain increasingly higher frequency components but are far from the high-frequency trigonometric basis functions.

## 5 RELATED LITERATURE

As mentioned before, the DIP paper (Ulyanov et al., 2018) was the first to show that over-parameterized convolutional networks enable solving denoising, inpainting, and super-resolution problems well even without any training data. Subsequently, the paper (Heckel & Hand, 2019) proposed a much simpler image generating network, termed the deep decoder. The papers (Veen et al., 2018; Heckel, 2019; Jagatap & Hegde, 2019; Bostan et al., 2020) have shown that the DIP and the deep decoder also enable solving or regularizing compressive sensing problems and other inverse problems.

Since the convolutional generators considered here are image-generating deep networks, our work is also related to methods that rely on trained deep image models. Deep learning based methods are either trained end-to-end for tasks ranging from compression (Toderici et al., 2016; Agustsson et al., 2017; Theis et al., 2017; Burger et al., 2012; Zhang et al., 2017) to denoising (Burger et al., 2012; Zhang et al., 2017), or are based on learning a generative image model (by training an autoencoder or GAN (Hinton & Salakhutdinov, 2006; Goodfellow et al., 2014)) and then using the resulting model to solve inverse problems such as compressed sensing (Bora et al., 2017; Hand & Voroninski, 2018), denoising (Heckel et al., 2018), or phase retrieval (Hand et al., 2018), by minimizing an associated loss. In contrast to the method studied here, where the optimization is over the weights

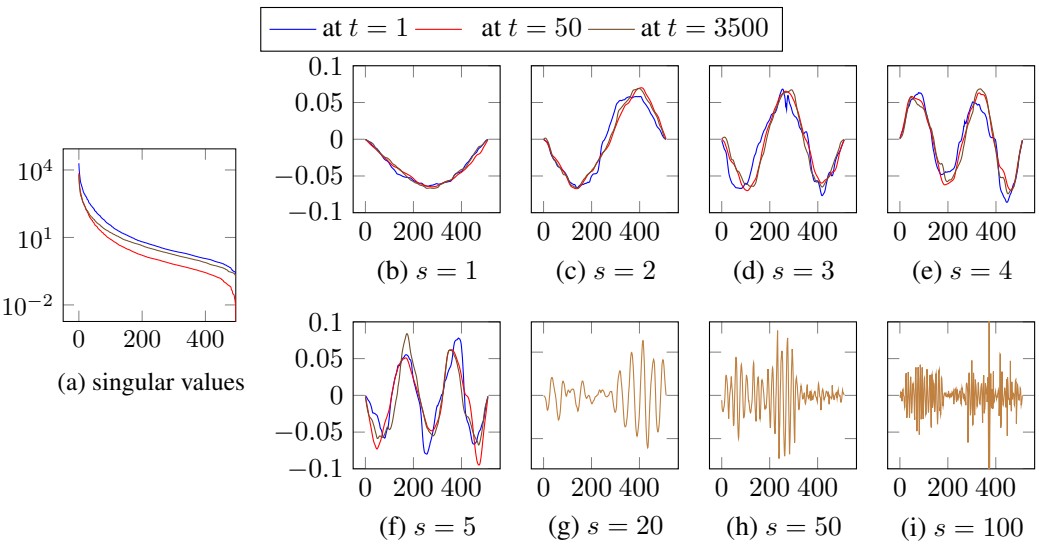

Figure 7: The Singular value distribution of the Jacobian of a four-layer deep decoder after $t = 50$ and $t = 3500$ iterations of gradient descent (panel **(a)**), along with the corresponding singular vectors/function **(b-i)**. The singular functions corresponding to the large singular vectors are close to the low-frequency Fourier modes and do not change significantly through training.

of the network, in all the aforementioned methods, the weights are adjusted only during training and then are fixed upon solving the inverse problem.

A large body of work focuses on understanding the optimization landscape of the simple nonlinearities or neural networks (Candes et al., 2015; Soltanolkotabi, 2017; Alon Brutzkus & Globerson, 2017; Zhong et al., 2017; Oymak, 2018; Fu et al., 2018; Tu et al., 2016) when the labels are created according to a planted model. Our proofs rely on showing that the dynamics of gradient descent on an over-parameterized network can be related to that of a linear network or a kernel problem. This proof technique has been utilized in a variety of recent publication (Soltanolkotabi et al., 2018; Venturi et al., 2019; Du et al., 2018; Oymak & Soltanolkotabi, 2019a;b; Arora et al., 2019; Oymak et al., 2019). Two recent publication have used this proof technique to show that functions are learned at different rates: Basri et al. (2019) have shown that functions of different frequencies are learned at different speeds, and Arora et al. (2019) has provided a theoretical explanation of the empirical observation that a simple 2-layer network fits random labels slower than actual labels in the context of classification. A recent publication (Li et al., 2019) focuses on demonstrating how early stopping leads to robust classification in the presence of label corruption under a cluster model for the input data. Neither of the aforementioned publication however, does address denoising in a regression setting or fitting convolutional generators of the form studied in this paper.

## CODE AND ACKNOWLEDGEMENTS

Code to reproduce the experiments is available at `https://github.com/MLI-lab/overparameterized_convolutional_generators`.

R. Heckel is partially supported by NSF award IIS-1816986, acknowledges support of the NVIDIA Corporation in form of a GPU, and would like to thank Tobit Klug for proofreading a previous version of this manuscript. M. Soltanolkotabi is supported by the Packard Fellowship in Science and Engineering, a Sloan Research Fellowship in Mathematics, an NSF-CAREER under award #1846369, the Air Force Office of Scientific Research Young Investigator Program (AFOSR-YIP) under award #FA9550-18-1-0078, an NSF-CIF award #1813877, DARPA under the Learning with Less Labels (LwLL) program, and a Google faculty research award.

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

# A  DENOISING PERFORMANCE OF UNTRAINED CONVOLUTIONAL GENERATORS

In this section, we provide further details on the denoising performance of deep neural networks. We compare the denoising performance of four methods: i) the BM3D algorithm (Dabov et al., 2007) as a standard baseline for denoising, ii) the deep image prior (Ulyanov et al., 2018), which is a U-net like encoder-decoder convolutional architecture applied with exactly the parameters as proposed for denoising in the paper (Ulyanov et al., 2018), and early stopped after 1900 iterations, again with the same stopping time as proposed in the original paper, iii) an under-parameterized deep decoder with five layers and $k = 128$ channels in each layers, trained for 3000 iterations (which is close to convergence), iv) an over-parameterized deep decoder with five layers and $k = 512$ channels in each layer, early stopped at 1500 iterations. We compared the performance of those four methods on denoising 100 randomly chosen images from the ImageNet validation set. The code to reproduce the results contains a list of the images we considered. Each image has $512 \times 512$ pixels and three color channels. We added the same random Gaussian noise to each color channel, because we are interested in evaluating the performance by imposing structural assumptions on the image. Table 1 below show that the overparameterized deep decoder with early stopping performs best for this task.

| | |
|---|---|
| Noise level | 20.60 |
| Deep decoder with k=512 and early stopping | 28.08dB |
| deep decoder with k=128 without early stopping | 27.84dB |
| DIP | 25.76 dB |
| BM3D | 25.52 dB |

Table 1:  Average performance for denoising images color image with the same Gaussian noise added to each color channel with BM3D and un-trained convolutional neural networks. The over-parameterized deep decoder performs best for this task.

# B  A NUMERICAL STUDY OF THE IMPLICIT BIAS OF CONVOLUTIONAL NETWORKS

In this section, we empirically demonstrate that convolutions with *fixed* convolutional kernels are critical for convolutional generators to fit natural images faster than noise. Towards this goal, we study numerically the following four closely related architectural choices, which differ in the upsampling/no-upsampling and convolutional operations which generate the activations in the $(i + 1)$-st layer, $\mathbf{B}_{i+1}$, from the activations in the $i$-th layer, $\mathbf{B}_i$:

  i) **Bilinear upsampling and linear combinations.** Layer $i + 1$ is obtained by linearly combining the channels of layer $i$ with learnable coefficients (i.e., performing one-times-one convolutions), followed by bi-linear upsampling. This is the deep decoder architecture from (Heckel et al., 2018).

 ii) **Fixed interpolation kernels and linear combinations.** Layer $i + 1$ is obtained by linearly combining the channels of layer $i$ with learnable coefficients followed by convolving each channel with the same 4x4 interpolation kernel that is used in the linear upsampling operator.

iii) **Parameterized convolutions:** Layer $i + 1$ is obtained from layer $i$ though a convolutional layer.

 iv) **Deconvolutional network:** Layer $i + 1$ is obtained from layer $i$ though a deconvolution layer. This is essentially the DC-GAN (Radford et al., 2015) generator architecture.

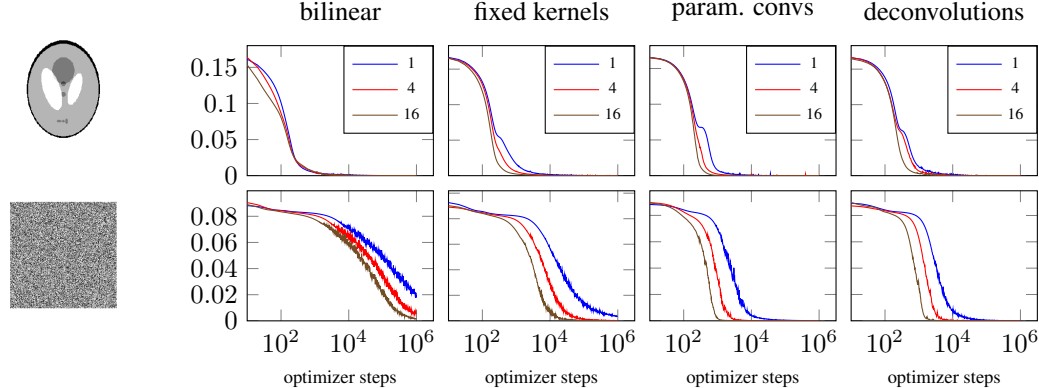

Figure 8: Fitting the phantom MRI and noise with different architectures of depth $d = 5$, for different number of over-parameterization factors (1,4, and 16). Gradient descent on convolutional generators involving fixed convolutional matrixes fit an image significantly faster than noise.

To emphasize that architectures i)-iv) are structurally very similar operations, we recall that each operation consists only of upsampling and convolutional operations. Let $\mathbf{T}(\mathbf{c})\colon \mathbb{R}^n \to \mathbb{R}^n$ be the convolutional operator with kernel $\mathbf{c}$, let $\mathbf{u}$ the linear upsampling kernel (equal to $\mathbf{u} = [0.5, 1, 0.5]$ in the one-dimensional case), and let $\mathbf{U}\colon \mathbb{R}^n \to \mathbb{R}^{2n}$ be an upsampling operator, that in the one dimensional case transforms $[x_1, x_2, \ldots, x_n]$ to $[x_1, 0, x_2, 0, \ldots, x_n, 0]$. In each of the architectures i)-iv), the $\ell$-th channel of layer $i + 1$ is obtained from the channels in the $i$-th layer as: $\mathbf{b}_{i+1,\ell} = \mathrm{ReLU}\left(\sum_{j=1}^{k} \mathbf{M}(\mathbf{c}_{ij\ell})\mathbf{b}_i\right)$, where the linear operator $\mathbf{M}$ is defined as follows for the four architectures

i) $\mathbf{M}(c) = c\mathbf{T}(\mathbf{u})\mathbf{U}$,  ii) $\mathbf{M}(c) = c\mathbf{T}(\mathbf{u})$,  iii) $\mathbf{M}(\mathbf{c}) = \mathbf{T}(\mathbf{c})$,  iv) $\mathbf{M}(\mathbf{c}) = \mathbf{T}(\mathbf{c})\mathbf{U}$.

The coefficients associated with the $i$-th layer are given by $\mathbf{C}_i = \{\mathbf{c}_{ij\ell}\}$, and all coefficients of the networks are $\mathbf{C} = \{\mathbf{c}_{ij\ell}\}$. Note that here, the coefficients or parameters of the networks are the weights and not the input to the network.

### B.1 DEMONSTRATING IMPLICIT BIAS OF CONVOLUTIONAL GENERATORS

We next show that convolutional generators with *fixed* convolutional operations fit natural or simple images significantly faster than complex images or noise. Throughout this section, for each image or signal $\mathbf{x}^*$ we fit weights by minimizing the loss

$$\mathcal{L}(\mathbf{C}) = \|G(\mathbf{C}) - \mathbf{x}^*\|_2^2$$

with respect to the network parameters $\mathbf{C}$ using plain gradient descent with a fixed stepsize.

In order to exemplify the effect, we fit the phantom MRI image as well as noise for each of the architectures above for a 5-layer network. We choose the number of channels, $k$, such that the over-parameterization factor (i.e., the ratio of number of parameters of the network over the output dimensionality) is $1, 4$, and $16$, respectively. The results in Figure 8 show that for architectures i) and ii) involving *fixed convolutional operations*, gradient descent requires more than one order of magnitude fewer iterations to obtain a good fit of the phantom MRI image relative to noise. For architectures iii) and iv), with trainable convolutional filters, we see a smaller effect, but the effect essentially vanishes when the network is highly over-parameterized.

This effect continues to exist for natural images in general, as demonstrated by Figure 10 which depicts the average and standard deviation of the loss curves of 100 randomly chosen images from the imagenet dataset.

We also note that the effect continues to exist in the sense that highly structured images with a large number of discontinuities are difficult to fit. An example is the checkerboard image in which each pixel alternates between 1 and 0; this image leads to the same loss curves as noise.

In our next experiment, we highlight that the distance between final and initial network weights is a key feature that determines the difference of fitting a natural image and noise. Towards this goal,

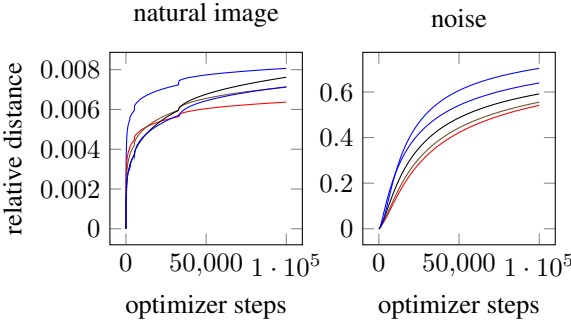

Figure 9: The relative distances of the weights in each layer from its random initialization. The weights need to change significantly more to fit the noise, compared to an image, thus a natural image lies closer to a random initialization than noise.

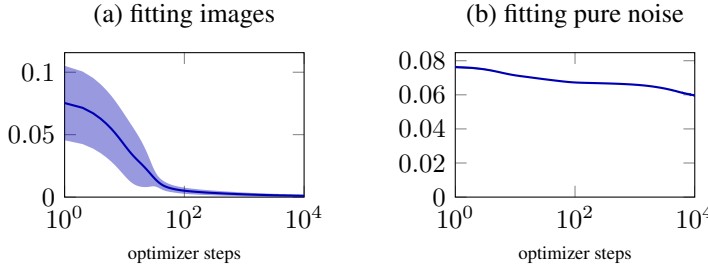

Figure 10: The loss curves for architecture i), a convolutional generator with linear upsampling operations, averaged over 100 $3 \times 512 \times 512$ (color) images from the Imagenet dataset. The error band is one standard deviation. Convolutional generators fit natural images significantly faster than noise.

we again fit the phantom MRI image and noise for the architecture i) and an over-parameterization factor of $4$ and record, for each layer $i$ the relative distance $\|\mathbf{C}_i^{(t)} - \mathbf{C}_i^{(0)}\| / \|\mathbf{C}_i^{(0)}\|$, where $\mathbf{C}_i^{(0)}$ are the weights at initialization (we initialize randomly), and $\mathbf{C}_i^{(t)}$ are the weights at the optimizer step $t$. The results, plotted in Figure 9, show that to fit the noise, the weights have to change significantly, while for fitting a natural image they only change slightly.

## C  THE SPECTRUM OF THE JACOBIAN OF THE DEEP DECODER AND DEEP IMAGE PRIOR

Our theoretical results predict that for over-parameterized networks, the parts of the signal that is aligned with the leading singular vectors of the Jacobian at initialization is fitted fastest. In this section we briefly show that natural images are much more aligned with the leading singular vectors than with Gaussian noise, which is equally aligned with each of the singular vectors.

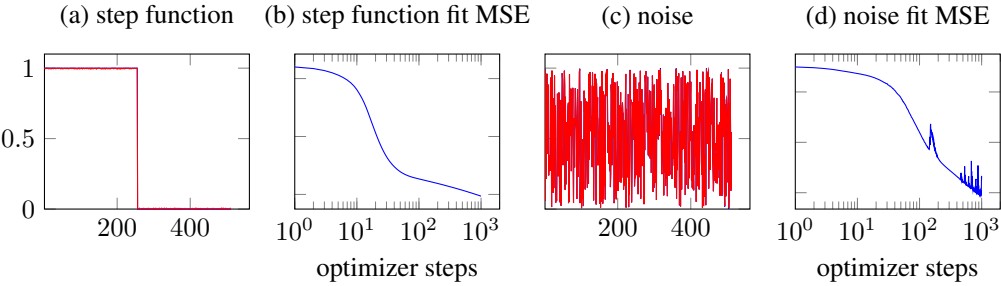

Figure 11: Fitting a step function and noise with a two-layer deep decoder: Even for a two-layer network, the simple image (step function) is fitted significantly faster than the noise.

(a) deep decoder                          (b) deep image prior

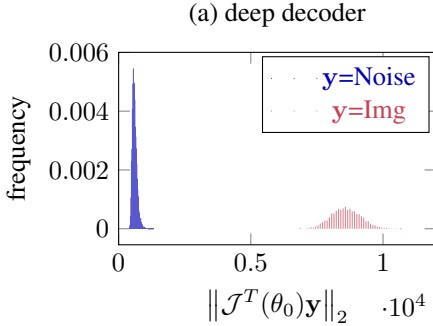 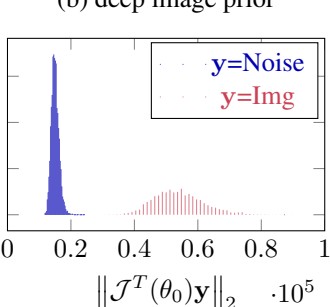

Figure 12: The distribution of the $\ell_2$-norm of the inner product of the Jacobian of a deep decoder (a) and a deep image prior (b) at a random initialization, with an image $\mathbf{y} = \mathbf{x}^*$ and noise $\mathbf{y} = \mathbf{z}$, both of equal norm. For both deep decoder and deep image prior, this quantity is significantly smaller for noise than for a natural image. Thus, a natural image is better aligned with the leading singular vectors of the Jacobian than noise. This demonstrates that the Jacobian of the networks is approximately low-rank, with natural images lying in the space spanned by the leading singularvectors.

Towards this goal, we compute the norm of the product of the Jacobian at a random initalization, $\mathbf{C}_0$, with a signal $\mathbf{y}$ as this measures the extent to which the signal is aligned with the leading singular vectors, due to

$$\left\| \mathcal{J}^T(\mathbf{C}_0)\mathbf{y} \right\|^2 = \left\| \mathbf{V}\Sigma\mathbf{W}^T\mathbf{y} \right\|^2 = \sum_i \sigma_i^2 \left\langle \mathbf{w}_i, \mathbf{y} \right\rangle^2,$$

where $\mathcal{J}(\mathbf{C}_0) = \mathbf{W}\Sigma\mathbf{V}^T$ is the singular value decomposition of the Jacobian.

Figure 12 depicts the distribution of the norm of the product of the Jacobian at initialization, $\mathcal{J}(\mathbf{C}_0)$, with an image $\mathbf{y}^*$ or noise $\mathbf{z}$ of equal norm ($\|\mathbf{y}^*\| = \|\mathbf{z}\|$). Since for both the deep decoder and the deep image prior, the norm of product of the Jacobian and the noise (i.e., $\left\| \mathcal{J}^T(\mathbf{C}_0)\mathbf{z} \right\|$) is significantly smaller than that with a natural image (i.e., $\left\| \mathcal{J}^T(\mathbf{C}_0)\mathbf{y}^* \right\|$), it follows that a structured image is much better aligned with the leading singular vectors of the Jacobian than the Gaussian noise, which is approximately equally aligned with any of the singular vectors. Thus, the Jacobian at random initialization has an approximate low-rank structure, with natural images lying in the space spanned by the leading singularvectors.

## D  PROOFS AND FORMAL STATEMENT OF RESULTS

The results stated in the main text are obtained from a slightly more general result which applies beyond convolutional networks. Consider neural network generators of the form

$$G(\mathbf{C}) = \text{ReLU}(\mathbf{UC})\mathbf{v}, \tag{8}$$

with $\mathbf{C} \in \mathbb{R}^{n \times k}$, and $\mathbf{U} \in \mathbb{R}^{n \times n}$ an arbitrary fixed matrix, and $\mathbf{v} \in \mathbb{R}^k$, with half of the entries equal to $+1/\sqrt{k}$ and the other half equal to $-1/\sqrt{k}$.

The (transposed) Jacobian of $\text{ReLU}(\mathbf{Uc})$ is $\mathbf{U}^T \text{diag}(\text{ReLU}'(\mathbf{Uc}))$. Thus the Jacobian of $G(\mathbf{C})$ is given by

$$\mathcal{J}^T(\mathbf{C}) = \begin{bmatrix} v_1 \mathbf{U}^T \text{diag}(\text{ReLU}'(\mathbf{Uc}_1)) \\ \vdots \\ v_k \mathbf{U}^T \text{diag}(\text{ReLU}'(\mathbf{Uc}_k)) \end{bmatrix} \in \mathbb{R}^{nk \times n}, \tag{9}$$

where $\text{ReLU}'$ is the derivative of the activation function. Next we define a notion of expected Jacobian. Towards this goal, we first define the matrix

$$\mathbf{\Sigma}(\mathbf{U}) \coloneqq \mathbb{E}\left[\mathcal{J}(\mathbf{C})\mathcal{J}^T(\mathbf{C})\right] \in \mathbb{R}^{n \times n}$$

associated with the generator $G(\mathbf{C}) = \text{ReLU}(\mathbf{UC})\mathbf{v}$. Here, expectation is over $\mathbf{C}$ with iid $\mathcal{N}(0, \omega^2)$ entries. Note that since the derivative of the ReLU non-linearity is invariant to the scaling of $\mathbf{C}$, so is the Jacobian $\mathcal{J}(\mathbf{C})$, but to be consistent throughout we take expectation over $\mathbf{C}$ with iid $\mathcal{N}(0, \omega^2)$ entries. Consider the eigenvalue decomposition of $\mathbf{\Sigma}(\mathbf{U})$ given by

$$\mathbf{\Sigma}(\mathbf{U}) = \sum_{i=1}^{n} \sigma_i^2 \mathbf{w}_i \mathbf{w}_i^T.$$

Our results depend on the largest and smallest eigenvalue of $\mathbf{\Sigma}(\mathbf{U})$, defined throughout as

$$\alpha = \sigma_n^2, \quad \beta = \sigma_1^2 = \|\mathbf{U}\|.$$

With these definitions in place we are ready to state our result about neural generators.

**Theorem 3.** *Consider a noisy signal $\mathbf{y} \in \mathbb{R}^n$ given by*

$$\mathbf{y} = \mathbf{x} + \mathbf{z},$$

*where $\mathbf{x} \in \mathbb{R}^n$ is assumed to lie in the signal subspace spanned by the $p$ leading singular vectors $\mathbf{w}_1, \ldots, \mathbf{w}_n$ of $\mathbf{\Sigma}(\mathbf{U})$, and $\mathbf{z} \in \mathbb{R}^n$ is an arbitrary noise vector. Suppose that the number of channels obeys*

$$k \geq cn\xi^{-8}\left(1 + \frac{\xi}{4}\frac{\alpha}{\beta}\eta T\beta^2\right)^2 \frac{\beta^{18}}{\alpha^{18}} \tag{10}$$

*where $c$ is a fixed numerical constant and $\xi$ an error tolerance parameter obeying $0 < \xi \leq 1/\sqrt{32 \log\left(\frac{2n}{\delta}\right)}$. Moreover, $T$ is the maximal number of gradient descent steps, and we assume it obeys $1 \leq T \leq \frac{2^5\beta^2}{\eta\xi^2\alpha^2}$. We fit the neural generator $G(\mathbf{C})$ to the noisy signal $\mathbf{y} \in \mathbb{R}^n$ by minimizing the loss*

$$\mathcal{L}(\mathbf{C}) = \frac{1}{2}\|G(\mathbf{C}) - \mathbf{y}\|_2^2 \tag{11}$$

*via running gradient descent starting from $\mathbf{C}_0$ with i.i.d. $\mathcal{N}(0, \omega^2)$ entries, $\omega = \frac{\|\mathbf{y}\|_2}{\sqrt{n}\beta}\xi\frac{\alpha^2}{\beta^2}$, and step size obeying $\eta \leq 1/\beta^2$. Then, with probability at least $1 - e^{-k^2} - \delta$, for all iterations $\tau \leq T$,*

$$\|\mathbf{x} - G(\mathbf{C}_\tau)\|_2 \leq (1 - \eta\sigma_p^2)^\tau \|\mathbf{x}\|_2 + \sqrt{\sum_{i=1}^{n}((1 - \eta\sigma_i^2)^\tau - 1)^2 \langle \mathbf{w}_i, \mathbf{z}\rangle^2} + \xi\|\mathbf{y}\|_2. \tag{12}$$

### D.1    PROOF OF THEOREM 2

Theorem 2 stated in the main text follow directly from Theorem 3 above as follows. We first note that for a convolutional generator (where $\mathbf{U}$ implements a convolution and thus is circulant) the eigenvectors of the matrix $\boldsymbol{\Sigma}(\mathbf{U})$ are given by the trigonometric basis functions per Definition 1 and the eigenvalues are the square of the entries of the dual kernel (Definition 2). To see this, we note that as detailed in Section H,

$$[\boldsymbol{\Sigma}(\mathbf{U})]_{ij} = \frac{1}{2}\left(1 - \cos^{-1}\left(\frac{\langle \mathbf{u}_i, \mathbf{u}_j\rangle}{\|\mathbf{u}_i\|\|\mathbf{u}_j\|}\right)/\pi\right)\langle \mathbf{u}_i, \mathbf{u}_j\rangle. \tag{13}$$

Because the matrix $\mathbf{U}$ implements a convolution with a kernel $\mathbf{u}$ that is equal to its first column, the matrix $\boldsymbol{\Sigma}(\mathbf{U})$ is again a circulant matrix and is also Hermitian. Thus, its spectrum is given by the Fourier transform of the first column of the circulant matrix, and its left-singular vectors are given by the trigonometric basis functions defined in equation (5) and depicted in Figure 3.

Furthermore, using the fact that the eigenvalues of a circulant matrix are given by its discrete Fourier transform we can substitute $\beta = \|\mathbf{U}\| = \|\mathbf{Fu}\|_\infty$ and $\alpha = \sigma_n > 0$. With the assumption $\tau \leq T = \frac{100}{\eta\sigma_p^2}$, and using that

$$1 + \frac{\xi}{4}\frac{\alpha}{\beta}T\beta^2 \leq 1 + 32\xi\frac{\beta}{\sigma_p} \leq 33,$$

where we used the assumption $\xi \leq \frac{\sigma_p}{\beta}$, the condition (10) is implied by the assumption $k \geq C_{\mathbf{u}}n/\xi^8$, with $C_{\mathbf{u}} \propto \frac{\beta^{18}}{\alpha^{18}}$ (see equation (6), where this assumption is made). This yields Theorem 2.

### D.2    PROOF OF THEOREM 1

Finally, we note that to obtain the simplified final expression in Theorem 1 from Theorem 2 we also used the fact that for a Gaussian vector $\mathbf{z}$, the vector $\mathbf{W}^T\mathbf{z}$ is also Gaussian. Furthermore, by the concentration of Lipschitz functions of Gaussians with high probability we have

$$\sum_{i=1}^n ((1 - \eta\sigma_i^2)^\tau - 1)^2 \langle \mathbf{w}_i, \mathbf{z}\rangle^2 \approx \mathbb{E}\left[\sum_{i=1}^n ((1 - \eta\sigma_i^2)^\tau - 1)^2 \langle \mathbf{w}_i, \mathbf{z}\rangle^2\right]$$

$$\overset{\text{(i)}}{=} \frac{\varsigma^2}{n}\sum_{i=1}^n ((1 - \eta\sigma_i^2)^\tau - 1)^2$$

$$\overset{\text{(ii)}}{\leq} \varsigma^2\frac{2p}{n}.$$

Here, equation (i) follows from $\langle \mathbf{w}_i, \mathbf{z}\rangle$ being zero mean Gaussian with variance $\varsigma^2/n$ (since $\mathbf{z} \sim \mathcal{N}(0, (\varsigma^2/n)\mathbf{I})$, and $\|\mathbf{w}_i\|_2 = 1$). Finally, (ii) follows by choosing the early stopping time so that it obeys $\tau \leq \log(1 - \sqrt{p/n})/\log(1 - \eta\sigma_{p+1}^2)$ which in turn implies that $(1 - \eta\sigma_i^2)^\tau \geq 1 - \sqrt{p/n}$, for all $i > p$, yielding that $((1 - \eta\sigma_i^2)^\tau - 1)^2 \leq p/n$, for all $i > p$.

## E    THE DYNAMICS OF LINEAR AND NONLINEAR LEAST-SQUARES

Theorem 3 builds on a more general result on the dynamics of a non-linear least squares problem which applies beyond convolutional networks, and that is stated and discussed in this section. Consider a nonlinear least-squares fitting problem of the form

$$\mathcal{L}(\boldsymbol{\theta}) = \frac{1}{2}\|f(\boldsymbol{\theta}) - \mathbf{y}\|_2^2.$$

Here, $f\colon \mathbb{R}^N \to \mathbb{R}^n$ is a non-linear model with parameters $\boldsymbol{\theta} \in \mathbb{R}^N$. To solve this problem, we run gradient descent with a fixed stepsize $\eta$, starting from an initial point $\boldsymbol{\theta}_0$, with updates of the form

$$\boldsymbol{\theta}_{\tau+1} = \boldsymbol{\theta}_\tau - \eta\nabla\mathcal{L}(\boldsymbol{\theta}_\tau) \quad \text{where} \quad \nabla\mathcal{L}(\boldsymbol{\theta}) = \mathcal{J}^T(\boldsymbol{\theta})(f(\boldsymbol{\theta}) - \mathbf{y}). \tag{14}$$

Here, $\mathcal{J}(\boldsymbol{\theta}) \in \mathbb{R}^{n \times N}$ is the Jacobian associated with the nonlinear map $f$ with entries given by $[\mathcal{J}(\boldsymbol{\theta})]_{i,j} = \frac{\partial f_i(\boldsymbol{\theta})}{\partial \boldsymbol{\theta}_j}$. In order to study the properties of the gradient descent iterates (14), we relate

the non-linear least squares problem to a linearized one in a ball around the initialization $\boldsymbol{\theta}_0$. We note that this general strategy has been utilized in a variety of recent publications (Du et al., 2018; Arora et al., 2019; Oymak & Soltanolkotabi, 2019b; Oymak et al., 2019). The associated linearized least-squares problem is defined as

$$\mathcal{L}_{\text{lin}}(\boldsymbol{\theta}) = \frac{1}{2}\|f(\boldsymbol{\theta}_0) + \mathbf{J}(\boldsymbol{\theta} - \boldsymbol{\theta}_0) - \mathbf{y}\|_2^2. \tag{15}$$

Here, $\mathbf{J} \in \mathbb{R}^{n \times p}$, refered to as the reference Jacobian, is a fixed matrix independent of $\boldsymbol{\theta}$ that approximates the Jacobian mapping at initialization, $\mathcal{J}(\boldsymbol{\theta}_0)$. Starting from the same initial point $\boldsymbol{\theta}_0$, the gradient descent updates of the linearized problem are

$$\widetilde{\boldsymbol{\theta}}_{\tau+1} = \widetilde{\boldsymbol{\theta}}_\tau - \eta\mathbf{J}^T\left(f(\boldsymbol{\theta}_0) + \mathbf{J}(\widetilde{\boldsymbol{\theta}}_\tau - \boldsymbol{\theta}_0) - \mathbf{y}\right)$$
$$= \widetilde{\boldsymbol{\theta}}_\tau - \eta\mathbf{J}^T\mathbf{J}\left(\widetilde{\boldsymbol{\theta}}_\tau - \boldsymbol{\theta}_0\right) - \eta\mathbf{J}^T\left(f(\boldsymbol{\theta}_0) - \mathbf{y}\right). \tag{16}$$

To show that the non-linear updates (14) are close to the linearized iterates (16), we make the following assumptions:

**Assumption 1** (Bounded spectrum). *We assume the singular values of the reference Jacobian obey*

$$\alpha \leq \sigma_n \leq \sigma_1 \leq \beta. \tag{17}$$

*Furthermore, we assume that the Jacobian mapping associated with the nonlinear model $f$ obeys*

$$\|\mathcal{J}(\boldsymbol{\theta})\| \leq \beta \quad \text{for all} \quad \boldsymbol{\theta} \in \mathbb{R}^N. \tag{18}$$

**Assumption 2** (Closeness of the reference and initialization Jacobians). *We assume the reference Jacobian and the Jacobian of the nonlinearity at initialization $\mathcal{J}(\boldsymbol{\theta}_0)$ are $\epsilon_0$-close in the sense that*

$$\|\mathcal{J}(\boldsymbol{\theta}_0) - \mathbf{J}\| \leq \epsilon_0. \tag{19}$$

**Assumption 3** (Bounded variation of Jacobian around initialization). *We assume that within a radius $R$ around the initialization, the Jacobian varies by no more than $\epsilon$ in the sense that*

$$\|\mathcal{J}(\boldsymbol{\theta}) - \mathcal{J}(\boldsymbol{\theta}_0)\| \leq \frac{\epsilon}{2}, \quad \text{for all} \quad \boldsymbol{\theta} \in \mathcal{B}_R(\boldsymbol{\theta}_0), \tag{20}$$

*where $\mathcal{B}_R(\boldsymbol{\theta}_0) := \{\boldsymbol{\theta}: \|\boldsymbol{\theta} - \boldsymbol{\theta}_0\| \leq R\}$ is the ball with radius $R$ around $\boldsymbol{\theta}_0$.*

Our first result shows that under these assumptions the nonlinear iterative updates (14) are intimately related to the linear iterative updates (16). Specifically, we show that the residuals associated with these two problems defined below

$$\text{nonlinear residual:} \quad \mathbf{r}_\tau := f(\boldsymbol{\theta}_\tau) - \mathbf{y} \tag{21}$$

$$\text{linear residual:} \quad \widetilde{\mathbf{r}}_\tau := \left(\mathbf{I} - \eta\mathbf{J}\mathbf{J}^T\right)^\tau \mathbf{r}_0, \tag{22}$$

are close in the proximity of the initialization.

**Theorem 4** (Closeness of linear and nonlinear least-squares problems). *Assume the Jacobian mapping $\mathcal{J}(\boldsymbol{\theta}) \in \mathbb{R}^{n \times N}$ associated with the function $f(\boldsymbol{\theta})$ obeys Assumptions 1, 2, and 3 around an initial point $\boldsymbol{\theta}_0 \in \mathbb{R}^N$ with respect to a reference Jacobian $\mathbf{J} \in \mathbb{R}^{n \times N}$ and with parameters $\alpha, \beta, \epsilon_0, \epsilon,$ and $R$. Furthermore, assume the radius $R$ is given by*

$$\frac{R}{2} := \left\|\mathbf{J}^\dagger\mathbf{r}_0\right\|_2 + \frac{1}{\alpha^2}(\epsilon_0 + \epsilon)\left(1 + 2\eta T\beta^2\right)\|\mathbf{r}_0\|_2, \tag{23}$$

*with $T$ a constant obeying $1 \leq T \leq \frac{1}{2\eta\epsilon^2}$, and $\mathbf{J}^\dagger$ the pseudo-inverse of $\mathbf{J}$. We run gradient descent with stepsize $\eta \leq \frac{1}{\beta^2}$ on the linear and non-linear least squares problem, starting from the same initialization $\boldsymbol{\theta}_0$. Then, for all $\tau \leq T$ the iterates of the original and the linearized problems and their corresponding residuals obey*

$$\|\mathbf{r}_\tau - \widetilde{\mathbf{r}}_\tau\|_2 \leq 2\frac{\beta}{\alpha^2}(\epsilon_0 + \epsilon)\|\mathbf{r}_0\|_2 \tag{24}$$

$$\left\|\boldsymbol{\theta}_\tau - \widetilde{\boldsymbol{\theta}}_\tau\right\|_2 \leq \frac{1}{\alpha^2}(\epsilon_0 + \epsilon)\left(1 + 2\eta\tau\beta^2\right)\|\mathbf{r}_0\|_2. \tag{25}$$

*Moreover, for all iterates $\tau \leq T$,*

$$\|\boldsymbol{\theta}_\tau - \boldsymbol{\theta}_0\|_2 \leq \frac{R}{2}. \tag{26}$$

The above theorem formalizes that in a (small) radius around the initialization, the non-linear problem behaves similarly as its linearization. Thus, to characterize the dynamics of the nonlinear problem, it suffices to characterize the dynamics of the linearized problem. This is the subject of our next theorem.

**Theorem 5.** *Consider a linear least squares problem* (15) *and let* $\mathbf{J} = \mathbf{W\Sigma V}^T \in \mathbb{R}^{n \times N} = \sum_{i=1}^{n} \sigma_i \mathbf{w}_i \mathbf{v}_i^T$ *be the singular value decomposition of the matrix* $\mathbf{J}$*. Then the residual* $\widetilde{\mathbf{r}}_\tau$ *after* $\tau$ *iterations of gradient descent with updates* (16) *is*

$$\widetilde{\mathbf{r}}_\tau = \sum_{i=1}^{n} \left(1 - \eta\sigma_i^2\right)^\tau \mathbf{w}_i \left\langle \mathbf{w}_i, \mathbf{r}_0 \right\rangle. \tag{27}$$

*Moreover, using a step size satisfying* $\eta \leq \frac{1}{\sigma_1^2}$*, the linearized iterates* (16) *obey*

$$\left\|\widetilde{\boldsymbol{\theta}}_\tau - \boldsymbol{\theta}_0\right\|_2^2 = \sum_{i=1}^{n} \left(\left\langle \mathbf{w}_i, \mathbf{r}_0 \right\rangle \frac{1 - (1 - \eta\sigma_i^2)^\tau}{\sigma_i}\right)^2. \tag{28}$$

In the next section we combine these two general theorems to provide guarantees for denoising using general neural networks.

### E.1 PROOF OF THEOREM 4 (CLOSENESS OF LINEAR AND NON-LINEAR LEAST-SQUARES)

The proof is by induction. We suppose the statement, in particular the bounds (24), (25), and (26) hold for iterations $t \leq \tau - 1$. We then show that they continue to hold for iteration $\tau$ in four steps. In Step I, we show that a weaker version of (26) holds, specifically that $\|\boldsymbol{\theta}_\tau - \boldsymbol{\theta}_0\|_2 \leq R$. Using this result, in Steps II and III we show that the bounds (24) and (25) hold, respectively. Finally, in Step IV we utilize Steps I-III to complete the proof of equation (26).

**Step I: Next iterate obeys** $\boldsymbol{\theta}_\tau \in \mathcal{B}_R(\boldsymbol{\theta}_0)$**.** To prove $\boldsymbol{\theta}_\tau \in \mathcal{B}_R(\boldsymbol{\theta}_0)$, first note that by the triangle inequality and the induction assumption (26) we have

$$\|\boldsymbol{\theta}_\tau - \boldsymbol{\theta}_0\|_2 \leq \|\boldsymbol{\theta}_\tau - \boldsymbol{\theta}_{\tau-1}\|_2 + \|\boldsymbol{\theta}_{\tau-1} - \boldsymbol{\theta}_0\|_2,$$
$$\leq \|\boldsymbol{\theta}_\tau - \boldsymbol{\theta}_{\tau-1}\|_2 + \frac{R}{2}.$$

So to prove $\|\boldsymbol{\theta}_\tau - \boldsymbol{\theta}_0\|_2 \leq R$ it suffices to show that $\|\boldsymbol{\theta}_\tau - \boldsymbol{\theta}_{\tau-1}\|_2 \leq R/2$. To this aim note that

$$\begin{aligned}
\frac{1}{\eta}\|\boldsymbol{\theta}_\tau - \boldsymbol{\theta}_{\tau-1}\|_2 &= \|\nabla\mathcal{L}(\boldsymbol{\theta}_{\tau-1})\|_2 \\
&= \left\|\mathcal{J}^T(\boldsymbol{\theta}_{\tau-1})\mathbf{r}_{\tau-1}\right\|_2 \\
&\leq \left\|\mathcal{J}^T(\boldsymbol{\theta}_{\tau-1})\widetilde{\mathbf{r}}_{\tau-1}\right\|_2 + \|\mathcal{J}(\boldsymbol{\theta}_{\tau-1})\| \left\|\mathbf{r}_{\tau-1} - \widetilde{\mathbf{r}}_{\tau-1}\right\|_2 \\
&\leq \left\|\mathbf{J}^T\widetilde{\mathbf{r}}_{\tau-1}\right\|_2 + \|\mathcal{J}(\boldsymbol{\theta}_{\tau-1}) - \mathbf{J}\| \left\|\widetilde{\mathbf{r}}_{\tau-1}\right\|_2 + \|\mathcal{J}(\boldsymbol{\theta}_{\tau-1})\| \left\|\mathbf{r}_{\tau-1} - \widetilde{\mathbf{r}}_{\tau-1}\right\|_2 \\
&\overset{(i)}{\leq} \left\|\mathbf{J}^T\widetilde{\mathbf{r}}_{\tau-1}\right\|_2 + (\epsilon + \epsilon_0)\|\widetilde{\mathbf{r}}_{\tau-1}\|_2 + \beta\|\mathbf{r}_{\tau-1} - \widetilde{\mathbf{r}}_{\tau-1}\|_2 \\
&\overset{(ii)}{\leq} \frac{1}{\eta}\left\|\mathbf{J}^\dagger\mathbf{r}_0\right\|_2 + (\epsilon + \epsilon_0)\|\mathbf{r}_0\|_2 + 2\frac{\beta^2}{\alpha^2}(\epsilon_0 + \epsilon)\|\mathbf{r}_0\|_2 \\
&\leq \frac{R}{2}.
\end{aligned}$$

In the above (i) follows from Assumptions 1, 2, and 3. For (ii), we bounded the last term with the induction hypothesis (24), the middle term with $\|\widetilde{\mathbf{r}}_{\tau-1}\|_2 \leq \|\mathbf{r}_0\|_2$, and the first term with the bound

$$
\begin{aligned}
\left\|\mathbf{J}^T \widetilde{\mathbf{r}}_{\tau-1}\right\|_2 &= \left\|\mathbf{J}^T (\mathbf{I} - \eta \mathbf{J}\mathbf{J}^T)^{\tau-1} \mathbf{r}_0\right\|_2 \\
&= \left\|\Sigma (\mathbf{I} - \eta \Sigma^2)^{\tau-1} \mathbf{W}^T \mathbf{r}_0\right\|_2 \\
&\leq \sqrt{\sum_{j=1}^{n} \sigma_j^2 \langle \mathbf{w}_j, \mathbf{r}_0 \rangle^2} \\
&\leq \beta^2 \sqrt{\sum_{j=1}^{n} \frac{1}{\sigma_j^2} \langle \mathbf{w}_j, \mathbf{r}_0 \rangle^2} \\
&= \beta^2 \left\|\mathbf{J}^\dagger \mathbf{r}_0\right\|_2 \\
&\leq \frac{1}{\eta} \left\|\mathbf{J}^\dagger \mathbf{r}_0\right\|_2.
\end{aligned}
$$

Finally, the last inequality follows by definition of $R$ in (23) together with the fact that $T \geq 1$.

**Step II: Original and linearized residuals are close:** In this step, we bound the deviation of the residuals of the original and linearized problem defined as

$$
\mathbf{e}_\tau := \mathbf{r}_\tau - \widetilde{\mathbf{r}}_\tau.
$$

This step relies on the following lemma, which is a variant of (Oymak et al., 2019, Lem. 6.7).

**Lemma 1** (Bound on growth of perturbations). *Suppose that Assumptions 1, 2, and 3 hold and that* $\boldsymbol{\theta}_\tau, \boldsymbol{\theta}_{\tau+1} \in \mathcal{B}_R(\boldsymbol{\theta}_0)$. *Then, provided the stepsize obeys* $\eta \leq 1/\beta^2$, *the deviation of the residuals obeys*

$$
\|\mathbf{e}_{\tau+1}\|_2 \leq \eta\beta \left(\epsilon_0 + \epsilon\right) \|\widetilde{\mathbf{r}}_\tau\|_2 + \left(1 + \eta\epsilon^2\right) \|\mathbf{e}_\tau\|_2. \tag{29}
$$

By the previous step, $\boldsymbol{\theta}_\tau, \boldsymbol{\theta}_{\tau+1} \in \mathcal{B}_R(\boldsymbol{\theta}_0)$. We next bound the two terms on the right hand side of (29). Regarding the first term, we note that an immediate consequence of Theorem 5 is the following bound on the residual of the linearized iterates:

$$
\|\widetilde{\mathbf{r}}_\tau\|_2 \leq \left(1 - \eta\alpha^2\right)^\tau \|\mathbf{r}_0\|_2. \tag{30}
$$

In order to bound the second term in (29), namely, $\|\mathbf{e}_\tau\|_2$, we used the following lemma, proven later in Section E.1.1.

**Lemma 2.** *Suppose that for positive scalars* $\alpha, \eta, \rho, \xi > 0$, $\eta \leq 1/\alpha^2$, *the sequences* $\widetilde{r}_\tau$ *and* $e_\tau$ *obey*

$$
\widetilde{r}_\tau \leq \left(1 - \eta\alpha^2\right)^\tau \rho \tag{31}
$$

$$
e_\tau \leq \left(1 + \eta\epsilon^2\right) e_{\tau-1} + \eta\xi \widetilde{r}_{\tau-1} \tag{32}
$$

*Then, for all* $\tau \leq \frac{1}{2\eta\epsilon^2}$, *we have that*

$$
e_\tau \leq 2\xi \frac{\rho}{\alpha^2}. \tag{33}
$$

With these lemmas in place we now have all the tools to prove that the original and linear residuals are close. In particular, from Step I, we know that $\boldsymbol{\theta}_\tau \in \mathcal{B}_R(\boldsymbol{\theta}_0)$ so that the assumptions of Lemma 1 are satisfied. Lemma 1 implies that the assumption (32) of Lemma 2 is satisfied with $\xi = \beta(\epsilon_0 + \epsilon)$ and the bound (30) implies that the assumption (31) of Lemma 2 is satisfied with $\rho = \|\mathbf{r}_0\|_2$. Thus, Lemma 2 implies that for all $\tau \leq \frac{1}{2\eta\epsilon^2}$

$$
\|\mathbf{e}_\tau\|_2 \leq 2\frac{\beta}{\alpha^2}(\epsilon_0 + \epsilon) \|\mathbf{r}_0\|_2. \tag{34}
$$

This concludes the proof of (24).

**Step III: Original and linearized parameters are close:** First note that by the triangle inequality and Assumptions 2 and 3 we have

$$\|\mathcal{J}(\boldsymbol{\theta}_\tau) - \mathbf{J}\| \leq \|\mathcal{J}(\boldsymbol{\theta}_\tau) - \mathcal{J}(\boldsymbol{\theta}_0)\| + \|\mathcal{J}(\boldsymbol{\theta}_0) - \mathbf{J}\| \leq \epsilon_0 + \epsilon. \tag{35}$$

The difference between the parameter of the original iterate $\boldsymbol{\theta}$ and the linearized iterate $\widetilde{\boldsymbol{\theta}}$ obey

$$\begin{aligned}
\frac{1}{\eta}\left\|\boldsymbol{\theta}_\tau - \widetilde{\boldsymbol{\theta}}_\tau\right\|_2 &= \left\|\sum_{t=0}^{\tau-1} \nabla\mathcal{L}(\boldsymbol{\theta}_t) - \nabla\mathcal{L}_{\mathrm{lin}}(\widetilde{\boldsymbol{\theta}}_t)\right\|_2 \\
&= \left\|\sum_{t=0}^{\tau-1} \mathcal{J}^T(\boldsymbol{\theta}_t)\mathbf{r}_t - \mathbf{J}^T\widetilde{\mathbf{r}}_t\right\|_2 \\
&\leq \sum_{t=0}^{\tau-1} \left\|(\mathcal{J}^T(\boldsymbol{\theta}_t) - \mathbf{J}^T)\widetilde{\mathbf{r}}_t\right\|_2 + \left\|\mathcal{J}^T(\boldsymbol{\theta}_t)(\mathbf{r}_t - \widetilde{\mathbf{r}}_t)\right\|_2 \\
&\stackrel{(i)}{\leq} \sum_{t=0}^{\tau-1} (\epsilon_0 + \epsilon)\|\widetilde{\mathbf{r}}_t\|_2 + \beta\|\mathbf{e}_t\|_2 \\
&\stackrel{(ii)}{\leq} (\epsilon_0 + \epsilon)\sum_{t=0}^{\tau-1} \left(1 - \eta\alpha^2\right)^{\tau-1}\|\mathbf{r}_0\|_2 + \beta\sum_{t=0}^{\tau-1}\|\mathbf{e}_t\|_2 \\
&= (\epsilon_0 + \epsilon)\frac{1 - (1 - \eta\alpha^2)^\tau}{\eta\alpha^2}\|\mathbf{r}_0\|_2 + \beta\sum_{t=0}^{\tau-1}\|\mathbf{e}_t\|_2 \\
&\stackrel{(iii)}{\leq} \frac{(\epsilon_0 + \epsilon)}{\eta\alpha^2}\|\mathbf{r}_0\|_2 + 2\tau\frac{\beta^2}{\alpha^2}(\epsilon_0 + \epsilon)\|\mathbf{r}_0\|_2 \\
&= \frac{1}{\eta\alpha^2}(\epsilon_0 + \epsilon)\left(1 + 2\eta\tau\beta^2\right)\|\mathbf{r}_0\|_2.
\end{aligned}$$

Here, inequality (i) follows from (35) and an application of Assumption 1, inequality (ii) from (30), inequality (iii) from $\eta \leq 1/\beta^2$ which implies $(1 - \eta\alpha^2) \geq 0$ and (34). This concludes the proof of the bound (25).

**Step IV: Completing the proof of** (26)**:** By the triangle inequality

$$\begin{aligned}
\|\boldsymbol{\theta}_\tau - \boldsymbol{\theta}_0\|_2 &\leq \left\|\widetilde{\boldsymbol{\theta}}_\tau - \boldsymbol{\theta}_0\right\|_2 + \left\|\boldsymbol{\theta}_\tau - \widetilde{\boldsymbol{\theta}}_\tau\right\|_2 \\
&\stackrel{(i)}{\leq} \left\|\mathbf{J}^\dagger\mathbf{r}_0\right\|_2 + \frac{1}{\alpha^2}(\epsilon_0 + \epsilon)\left(1 + 2\eta\tau\beta^2\right)\|\mathbf{r}_0\|_2 \\
&\stackrel{(ii)}{\leq} R/2,
\end{aligned}$$

where inequality (i) follows from the bound (25), which we just proved, and the fact that, from equation (36) in Theorem 5,

$$\begin{aligned}
\left\|\widetilde{\boldsymbol{\theta}}_\tau - \boldsymbol{\theta}_0\right\|_2^2 &= \sum_{i=1}^{n} \langle\mathbf{w}_i, \mathbf{r}_0\rangle^2 \frac{(1 - (1 - \eta\sigma_i^2)^\tau)^2}{\sigma_i^2} \\
&\leq \sum_{i=1}^{n} \langle\mathbf{w}_i, \mathbf{r}_0\rangle^2 / \sigma_i^2 \\
&= \left\|\mathbf{J}^\dagger\mathbf{r}_0\right\|^2.
\end{aligned}$$

Finally, inequality (ii) follows from the definition of $R$ in equation (23) along with $\tau \leq T$, by assumption.

### E.1.1 PROOF OF LEMMA 2

We prove the result by induction. Assume equation (33) holds true for some $\tau$. We prove that then it also holds true for $\tau + 1$. By the two assumptions in the lemma,

$$
\begin{aligned}
e_{\tau+1} - e_\tau &\le \eta\epsilon^2 e_\tau + \eta\xi\widetilde{r}_\tau \\
&\le \eta\epsilon^2 e_\tau + \eta\xi(1 - \eta\alpha^2)^\tau \rho \\
&\overset{(i)}{\le} \eta\xi\left(\epsilon^2\frac{2\rho}{\alpha^2} + (1 - \eta\alpha^2)^\tau \rho\right),
\end{aligned}
$$

where (i) follows from the induction assumption (33). Summing up the difference of the errors gives

$$
\begin{aligned}
\frac{e_\tau}{\xi} &= \sum_{t=0}^{\tau-1}\frac{(e_{t+1} - e_t)}{\xi} \\
&\le \tau\eta\epsilon^2\frac{2\rho}{\alpha^2} + \eta\rho\sum_{t=0}^{\tau-1}(1 - \eta\alpha^2)^t \\
&= \tau\eta\epsilon^2\frac{2\rho}{\alpha^2} + \eta\rho\frac{1 - (1 - \eta\alpha^2)^\tau}{\eta\alpha^2} \\
&\le \frac{\rho}{\alpha^2}(\eta 2\tau\epsilon^2 + 1) \\
&\le 2\frac{\rho}{\alpha^2},
\end{aligned}
$$

where the last inequality follows from the assumption that $\tau \le \frac{1}{2\eta\epsilon^2}$.

### E.2 PROOF OF THEOREM 5

The proof of identity (27) is equivalent to the proof of Proposition (1). Regarding inequality (28), note that

$$
\widetilde{\boldsymbol{\theta}}_\tau - \widetilde{\boldsymbol{\theta}}_0 = -\eta\sum_{t=0}^{\tau-1}\nabla\mathcal{L}_{\mathrm{lin}}(\widetilde{\boldsymbol{\theta}}_t) = -\eta\sum_{t=0}^{\tau-1}\mathbf{J}^T\widetilde{\mathbf{r}}_t = -\eta\mathbf{V}\left(\sum_{t=0}^{\tau-1}\boldsymbol{\Sigma}\left(\mathbf{I} - \eta\boldsymbol{\Sigma}^2\right)^t\right)\begin{bmatrix}\langle\mathbf{w}_1,\mathbf{r}_0\rangle \\ \vdots \\ \langle\mathbf{w}_n,\mathbf{r}_0\rangle\end{bmatrix}.
$$

Thus

$$
\left\langle\mathbf{v}_i,\widetilde{\boldsymbol{\theta}}_\tau - \widetilde{\boldsymbol{\theta}}_0\right\rangle = -\eta\sigma_i\langle\mathbf{w}_i,\mathbf{r}_0\rangle\left(\sum_{t=0}^{\tau-1}(1 - \eta\sigma_i^2)^t\right) = -\eta\sigma_i\langle\mathbf{w}_i,\mathbf{r}_0\rangle\frac{1 - (1 - \eta\sigma_i^2)^\tau}{\eta\sigma_i^2}. \tag{36}
$$

This in turn implies that

$$
\left\|\widetilde{\boldsymbol{\theta}}_\tau - \widetilde{\boldsymbol{\theta}}_0\right\|_2^2 = \sum_{i=1}^n\left\langle\mathbf{v}_i,\widetilde{\boldsymbol{\theta}}_\tau - \widetilde{\boldsymbol{\theta}}_0\right\rangle^2 = \sum_{i=1}^n\left(\langle\mathbf{w}_i,\mathbf{r}_0\rangle\frac{1 - (1 - \eta\sigma_i^2)^\tau}{\sigma_i}\right)^2.
$$

## F PROOFS FOR NEURAL NETWORK GENERATORS (PROOF OF THEOREM 3)

The proof of Theorem 3 relies on the fact that, in the overparameterized regime, the non-linear least squares problem is well approximated by an associated linearized least squares problem. Studying the associated linear problem enables us to prove the result.

We apply Theorem 4, which ensures that the associated linear problem is a good approximation of the non-linear least squarest problem, with the nonlinearity $G(\mathbf{C}) = \mathrm{ReLU}(\mathbf{UC})\mathbf{v}$ and with the parameter given by $\boldsymbol{\theta} = \mathbf{C}$. Recall that $\mathbf{v}$ is a fixed vector with half of the entries $1/\sqrt{k}$, and the other half $-1/\sqrt{k}$.

As the reference Jacobian in the associated linear problem, we choose a matrix $\mathbf{J} \in \mathbb{R}^{n\times nk}$, that is very close to the original Jacobian at initialization, $\mathcal{J}(\mathbf{C}_0)$, and that obeys $\mathbf{JJ}^T =$

$\mathbb{E}\left[\mathcal{J}(\mathbf{C})\mathcal{J}^T(\mathbf{C})\right]$. The concrete choice of $\mathbf{J}$ is discussed in the next paragraph. We apply Theorem 4 with the following choices of parameters:

$$\alpha = \sigma_n\left(\mathbf{\Sigma}(\mathbf{U})\right), \quad \beta = \|\mathbf{U}\|, \quad \epsilon_0 = \beta\left(4\frac{\log(\frac{2n}{\delta})}{k}\right)^{1/4}, \quad \epsilon = \frac{\xi}{8}\beta\frac{\alpha^2}{\beta^2}, \quad \omega = \frac{\|\mathbf{y}\|_2}{\sqrt{n}\beta}\xi\frac{\alpha^2}{\beta^2}.$$

A few comments on the intuition behind those choices: As shown below, $\alpha$ and $\beta$ are chosen so that Assumption 1 is satisfied. Regarding the choices of $\epsilon, \epsilon_0$, and $\omega$, recall that the difference between the linear and non-linear residual is bounded by $2\frac{\beta}{\alpha^2}(\epsilon_0 + \epsilon)\|\mathbf{r}_0\|_2$ (by Theorem 4). We want this error to be bounded by $\xi\|\mathbf{y}\|_2$. This bound is guaranteed by our choice of $\epsilon$, by choosing $\epsilon_0$ such that it obeys $\epsilon_0 \le \epsilon$, and by choosing $\omega$ so that the initial residual is bounded by $\|\mathbf{r}_0\|_2 \le 2\|\mathbf{y}\|_2$. Note that $\epsilon_0 \le \epsilon$ holds by assumption (10).

We next verify by applying a series of Lemmas proven in Appendix I that the conditions of Theorem 4 are satisfied (specifically, Assumptions 1, 2, 3 on the Jacobians of the non-linear map and the associated linearized map, to be bounded and sufficiently close to each other).

**Choice of reference Jacobian and verification of assumption 2:** We chose the reference Jacobian $\mathbf{J}$ so that it is $\epsilon_0$ close to the random initialization $\mathcal{J}(\mathbf{C}_0)$ (with high probability), and thus satisfies assumption 2. Towards this goal, we require the following two lemmas.

**Lemma 3** (Concentration lemma). *Consider $G(\mathbf{C}) = \text{ReLU}(\mathbf{U}\mathbf{C})\mathbf{v}$ with $\mathbf{v} \in \mathbb{R}^k$ and $\mathbf{U} \in \mathbb{R}^{n \times k}$ and associated Jacobian $\mathcal{J}(\mathbf{C})$ (9). Let $\mathbf{C} \in \mathbb{R}^{n \times k}$ be generated at random with i.i.d. $\mathcal{N}(0, \omega^2)$ entries. Then,*

$$\left\|\mathcal{J}(\mathbf{C})\mathcal{J}^T(\mathbf{C}) - \mathbf{\Sigma}(\mathbf{U})\right\| \le \|\mathbf{U}\|^2\sqrt{\log\left(\frac{2n}{\delta}\right)\sum_{\ell=1}^k v_\ell^4},$$

*holds with probability at least $1 - \delta$.*

To see that Lemma 3 implies the condition (19), we use the following lemma.

**Lemma 4** ((Oymak et al., 2019, Lem. 6.4)). *Let $\mathbf{X} \in \mathbb{R}^{n \times N}$, $N \ge n$ and let $\mathbf{\Sigma}$ be $n \times n$ psd matrix obeying $\left\|\mathbf{X}\mathbf{X}^T - \mathbf{\Sigma}\right\| \le \epsilon_0^2/4$, for a scalar $\epsilon \ge 0$. Then there exists a matrix $\mathbf{J} \in \mathbb{R}^{n \times N}$ obeying $\mathbf{\Sigma} = \mathbf{J}\mathbf{J}^T$ such that*

$$\|\mathbf{J} - \mathbf{X}\| \le \epsilon_0.$$

In order to verify the condition (19), note that using the fact that $\sum_\ell^k v_\ell^4 = \frac{1}{k}$ by Lemma 3, with probability at least $1 - \delta$, we have

$$\left\|\mathcal{J}(\mathbf{C}_0)\mathcal{J}^T(\mathbf{C}_0) - \mathbf{\Sigma}(\mathbf{U})\right\| \le \|\mathbf{U}\|^2\sqrt{\frac{\log\left(\frac{2n}{\delta}\right)}{k}} = (\epsilon_0/2)^2, \tag{37}$$

Combining inequality (37) with Lemma 4 (applied with $\mathbf{X} = \mathcal{J}(\mathbf{C}_0)$), it follows that condition (19) holds for the chosen value of $\epsilon_0$, concluding the proof of Assumption 2 being satisfied. This proof specifies the reference Jacobian $\mathbf{J}$ as the matrix that is $\epsilon_0$-close to $\mathcal{J}(\mathbf{C}_0)$, and that exists by Lemma 4.

**Verifying Assumption 1:** To verify Assumption 1, note that by definition $\mathbf{J}\mathbf{J}^T = \mathbf{\Sigma}(\mathbf{U})$ thus trivially $\sigma_n\left(\mathbf{\Sigma}(\mathbf{U})\right) \ge \alpha$ holds. Furthermore, Lemma 5 below combined with the fact that $\|\mathbf{v}\|_2 = 1$ implies that $\|\mathbf{J}\| \le \beta$ and $\|\mathcal{J}(\mathbf{C})\| \le \beta$ for all $\mathbf{C}$. This completes the verification of Assumption 1. It remains to show that the Jacobian has bounded spectrum:

**Lemma 5** (Spectral norm of Jacobian). *Consider $G(\mathbf{C}) = \text{ReLU}(\mathbf{U}\mathbf{C})\mathbf{v}$ with $\mathbf{v} \in \mathbb{R}^k$ and $\mathbf{U} \in \mathbb{R}^{n \times k}$ and associated Jacobian $\mathcal{J}(\mathbf{C})$ (9), and let $\mathbf{J}$ be any matrix obeying $\mathbf{J}\mathbf{J}^T = \mathbb{E}\left[\mathcal{J}(\mathbf{C})\mathcal{J}^T(\mathbf{C})\right]$, where the expectation is over a matrix $\mathbf{C}$ with iid $\mathcal{N}(0, \omega^2)$ entries. Then*

$$\|\mathcal{J}(\mathbf{C})\| \le \|\mathbf{v}\|_2\|\mathbf{U}\| \quad \text{and} \quad \|\mathbf{J}\| \le \|\mathbf{v}\|_2\|\mathbf{U}\|.$$

**Bound on initial residual:** For what follows, we require a bound on the initial residual. To prove this bound, we apply the following lemma.

**Lemma 6** (Initial residual). *Consider $G(\mathbf{C}) = \mathrm{ReLU}(\mathbf{U}\mathbf{C})\mathbf{v}$, and let $\mathbf{C} \in \mathbb{R}^{n \times k}$ be generated at random with i.i.d. $\mathcal{N}(0, \omega^2)$ entries. Suppose half of the entries of $\mathbf{v}$ are $\nu/\sqrt{k}$ and the other half are $-\nu/\sqrt{k}$, for some constant $\nu > 0$. Then, with probability at least $1 - \delta$,*

$$\|G(\mathbf{C})\|_2 \leq \nu\omega\sqrt{8\log(2n/\delta)} \|\mathbf{U}\|_F.$$

Now, the initial residual can be upper bounded as

$$\|\mathbf{r}_0\|_2 \leq \|\mathbf{y}\|_2 + \|G(\mathbf{C}_0)\|_2 \leq \frac{3}{2}\|\mathbf{y}\|_2, \tag{38}$$

where we used that, by Lemma 6,

$$\|G(\mathbf{C}_0)\|_2 \leq \omega\sqrt{8\log(2n/\delta)} \|\mathbf{U}\|_F \leq \|\mathbf{y}\|_2 \xi \frac{\alpha^2}{\beta^2}\sqrt{8\log(2n/\delta)} \leq \frac{1}{2}\|\mathbf{y}\|_2. \tag{39}$$

Here, the second inequality follows from $\|\mathbf{U}\|_F \leq \sqrt{n}\|\mathbf{U}\|$ and our choice of the parameter $\omega$, and the last inequality follows from $\xi \leq 1/\sqrt{32\log(2n/\delta)}$ and $\alpha/\beta \leq 1$.

**Verifying Assumption 3:** Verification of the assumption requires us to control the perturbation of the Jacobian matrix around a random initialization.

**Lemma 7** (Jacobian perturbation around initialization). *Let $\mathbf{C}_0$ be a matrix with i.i.d. $N(0, \omega^2)$ entries. Then, for all $\mathbf{C}$ obeying*

$$\|\mathbf{C} - \mathbf{C}_0\| \leq \omega\widetilde{R} \quad with \quad \widetilde{R} \leq \frac{1}{2}\sqrt{k},$$

*the Jacobian mapping (9) associated with the generator $G(\mathbf{C}) = \mathrm{ReLU}(\mathbf{U}\mathbf{C})\mathbf{v}$ obeys*

$$\|\mathcal{J}(\mathbf{C}) - \mathcal{J}(\mathbf{C}_0)\| \leq \|\mathbf{v}\|_\infty 2(k\widetilde{R})^{1/3} \|\mathbf{U}\|,$$

*with probability at least $1 - ne^{-\frac{1}{2}\widetilde{R}^{4/3}k^{7/3}}$.*

In order to verify Assumption 3, first note that the radius in the theorem, defined in equation (23), obeys

$$\begin{aligned}
R &= 2\|\mathbf{J}^\dagger\mathbf{r}_0\|_2 + \frac{2}{\alpha^2}(\epsilon_0 + \epsilon)\left(1 + 2\eta T\beta^2\right)\|\mathbf{r}_0\|_2 \\
&\overset{(i)}{\leq} \left(\frac{2}{\alpha} + \frac{2}{\alpha^2}(\epsilon_0 + \epsilon)\left(1 + 2\eta T\beta^2\right)\right)\|\mathbf{r}_0\|_2 \\
&\overset{(ii)}{\leq} \frac{2}{\alpha}\left(1 + \frac{2\epsilon}{\alpha}\left(1 + 2\eta T\beta^2\right)\right)2\omega\sqrt{n}\xi^{-1}\beta\frac{\beta^2}{\alpha^2} = 4\left(1 + \frac{2\epsilon}{\alpha}\left(1 + 2\eta T\beta^2\right)\right)\omega\sqrt{n}\xi^{-1}\frac{\beta^3}{\alpha^3} \\
&\overset{(iii)}{\leq} 4\omega\left(2 + \frac{\xi}{2}\frac{\alpha}{\beta}\eta T\beta^2\right)\sqrt{n}\xi^{-1}\frac{\beta^3}{\alpha^3} \\
&\overset{(iv)}{\leq} \omega\left(\frac{\xi}{16}\frac{\alpha^2}{\beta^2}\right)^3\sqrt{k} \\
&:= \omega\widetilde{R}.
\end{aligned}$$

Here, (i) follows from the fact that $\left\|\mathbf{J}_\mathcal{S}^\dagger\mathbf{r}_0\right\|_2 \leq \frac{1}{\alpha}\|\mathbf{r}_0\|_2$, (ii) from the bound on the initial residual (38), using that $\epsilon_0 \leq \epsilon$, as well as our choice of $\omega$. Moreover (iii) follows from the definition of $\epsilon$ and (iv) from the lower bound on $k$ in assumption (10).

Note that since $\frac{\xi^2}{16}\frac{\alpha^2}{\beta^2} \leq \frac{1}{2}$ for this choice of radius $\tilde{R}$, by Lemma 7 we have, with $\|\mathbf{v}\|_\infty = 1/\sqrt{k}$, that

$$\|\mathcal{J}(\mathbf{C}) - \mathcal{J}(\mathbf{C}_0)\| \leq \|\mathbf{v}\|_\infty 2(k\widetilde{R})^{1/3} \|\mathbf{U}\| = \frac{1}{\sqrt{k}}2\frac{\xi}{16}\frac{\alpha^2}{\beta^2}(k \cdot k^{1/2})^{1/3}\beta = \frac{\xi}{8}\frac{\alpha^2}{\beta} = \epsilon.$$

holds with probability at least

$$1 - ne^{-\frac{1}{2}\tilde{R}k^{7/3}} = 1 - ne^{-2^{-17}\xi^4\frac{\alpha^8}{\beta^8}k^3} \overset{(i)}{\geq} 1 - \delta,$$

where in (i) we used (10) together with $\xi \leq \sqrt{8\log\left(\frac{2n}{\delta}\right)}$. Therefore, Assumption 3 holds with high probability by our choice of $\epsilon = \frac{\xi}{8}\frac{\alpha^2}{\beta}$.

**Verifying the bound on number of iterations:**  Finally we note that the constraint on the number of iterations, $T$, $T \leq \frac{1}{2\eta\epsilon^2}$, from Theorem 4 is satisfied under the number of constraints of Theorem 3 by $\frac{1}{2\eta\epsilon^2} = \frac{2^5\beta^2}{\eta\xi^2\alpha^4}$.

**Concluding the proof of Theorem 3:**  Now that we have verified the conditions of Theorem 4 we can apply the theorem. This allows us to conclude that

$$
\begin{aligned}
\|G(\mathbf{C}_\tau) - \mathbf{x}\|_2 &= \|G(\mathbf{C}_\tau) + \mathbf{z} - \mathbf{y}\|_2 \\
&= \|\mathbf{r}_\tau + \mathbf{z}\|_2 \\
&= \|\widetilde{\mathbf{r}}_\tau + \mathbf{z} + \mathbf{r}_\tau - \widetilde{\mathbf{r}}_\tau\|_2 \\
&\overset{(i)}{\leq} \|\widetilde{\mathbf{r}}_\tau + \mathbf{z}\|_2 + \|\mathbf{r}_\tau - \widetilde{\mathbf{r}}_\tau\|_2 \\
&\overset{(ii)}{\leq} \|\widetilde{\mathbf{r}}_\tau + \mathbf{z}\|_2 + \xi\|\mathbf{y}\|_2 \\
&\overset{(iii)}{=} \left\|\mathbf{W}\left(\mathbf{I} - \eta\mathbf{\Sigma}^2\right)^\tau \mathbf{W}^T\mathbf{r}_0 + \mathbf{z}\right\|_2 + \frac{1}{2}\xi\|\mathbf{y}\|_2 \\
&\overset{(iv)}{=} \left\|\mathbf{W}\left(\mathbf{I} - \eta\mathbf{\Sigma}^2\right)^\tau \mathbf{W}^T(G(\mathbf{C}_0) - \mathbf{x}) - \left(\mathbf{W}\left(\mathbf{I} - \eta\mathbf{\Sigma}^2\right)^\tau \mathbf{W}^T - \mathbf{I}\right)\mathbf{z}\right\|_2 + \frac{1}{2}\xi\|\mathbf{r}_0\|_2 \\
&\overset{(v)}{\leq} \left\|\left(\mathbf{I} - \eta\mathbf{\Sigma}^2\right)^\tau \mathbf{W}^T\mathbf{x}\right\|_2 + \left\|\left(\left(\mathbf{I} - \eta\mathbf{\Sigma}^2\right)^\tau - \mathbf{I}\right)\mathbf{W}^T\mathbf{z}\right\|_2 + \|G(\mathbf{C}_0)\|_2 + \frac{1}{2}\xi\|\mathbf{y}\|_2 \\
&\overset{(vi)}{\leq} (1 - \eta\sigma_p^2)^\tau\|\mathbf{x}\|_2 + \sqrt{\sum_{i=1}^{n}((1 - \eta\sigma_i^2)^\tau - 1)^2\langle\mathbf{w}_i, \mathbf{z}\rangle^2} + \xi\|\mathbf{y}\|_2.
\end{aligned}
$$

Here, (i) follows from the triangular inequality, (ii) from Theorem 4 equation (24) combined with our choice for $\epsilon$ and $\|\mathbf{r}_0\|_2 \leq \frac{1}{2}\|\mathbf{y}\|_2$, shown above. Moreover, (iii) follows from Theorem 5, (iv) from $\mathbf{r}_0 = G(\mathbf{C}_0) - \mathbf{y} = G(\mathbf{C}_0) - \mathbf{x} - \mathbf{z}$, (v) from the triangular inequality. Finally, for (vi) we used the fact that $\mathbf{x} \in \text{span}\{\mathbf{w}_1, \mathbf{w}_2, \ldots, \mathbf{w}_p\}$ combined with the fact that $\left\|\mathbf{W}\left(\mathbf{I} - \eta\mathbf{\Sigma}^2\right)^\tau \mathbf{W}^T\right\| \leq 1$, as well as using the bound $\|G(\mathbf{C}_0)\|_2 \leq \frac{1}{2}\|\mathbf{y}\|_2$ proven in (39). This proves the final bound (12), as desired.

## G  PROOF OF PROPOSITION 1 AND EQUATION (2)

The residual of gradient descent at iteration $\tau$ is

$$
\begin{aligned}
\mathbf{r}_\tau &= \mathbf{y} - \mathbf{J}\mathbf{c}_\tau \\
&= \mathbf{y} - \mathbf{J}(\mathbf{c}_{\tau-1} - \eta\mathbf{J}^T(\mathbf{J}\mathbf{c}_{\tau-1} - \mathbf{y})) \\
&= (\mathbf{I} - \eta\mathbf{J}\mathbf{J}^T)(\mathbf{y} - \mathbf{J}\mathbf{c}_{\tau-1}) \\
&= (\mathbf{I} - \eta\mathbf{J}\mathbf{J}^T)^\tau(\mathbf{y} - \mathbf{J}\mathbf{c}_0) \\
&= (\mathbf{I} - \eta\mathbf{J}\mathbf{J}^T)^\tau\mathbf{y} \\
&= (\mathbf{I} - \eta\mathbf{W}\mathbf{\Sigma}^2\mathbf{W}^T)^\tau\mathbf{y}
\end{aligned}
$$

where we used that $\mathbf{c}_0 = 0$ and the SVD $\mathbf{J} = \mathbf{W}\mathbf{\Sigma}\mathbf{V}^T$. Expanding $\mathbf{y}$ in terms of the singular vectors $\mathbf{w}_i$ (i.e., the columns of $\mathbf{W}$), as $\mathbf{y} = \sum_i \mathbf{w}_i\langle\mathbf{w}_i, \mathbf{y}\rangle$, and noting that $(\mathbf{I} - \eta\mathbf{W}\mathbf{\Sigma}^2\mathbf{W}^T)^\tau = \sum_i(1 - \eta\sigma_i^2)^\tau\mathbf{w}_i\mathbf{w}_i^T$ we get

$$\mathbf{r}_\tau = \sum_i(1 - \eta\sigma_i^2)^\tau\mathbf{w}_i\langle\mathbf{w}_i, \mathbf{y}\rangle,$$

as desired.

**Proof of equation** (2): By proposition 1 and using that $\mathbf{x}$ and lies in the signal subspace

$$\mathbf{x} - \mathbf{J}\mathbf{c}_\tau = \sum_{i=1}^p \mathbf{w}_i (1 - \eta\sigma_i^2)^\tau \langle \mathbf{w}_i, \mathbf{x} \rangle + \sum_{i=1}^n \mathbf{w}_i ((1 - \eta\sigma_i^2)^\tau - 1) \langle \mathbf{w}_i, \mathbf{z} \rangle.$$

By the triangle inequality,

$$\|\mathbf{x} - \mathbf{J}\mathbf{c}_\tau\|_2 \le \left\| \sum_{i=1}^p \mathbf{w}_i (1 - \eta\sigma_i^2)^\tau \langle \mathbf{w}_i, \mathbf{x} \rangle \right\|_2 + \left\| \sum_{i=1}^n \mathbf{w}_i ((1 - \eta\sigma_i^2)^\tau - 1) \langle \mathbf{w}_i, \mathbf{z} \rangle \right\|_2$$

$$\le (1 - \eta\sigma_p^2)^\tau \|\mathbf{x}\|_2 + \sqrt{\sum_{i=1}^n ((1 - \eta\sigma_i^2)^\tau - 1)^2 \langle \mathbf{w}_i, \mathbf{z} \rangle^2},$$

where the second inequality follows by using orthogonality of the $\mathbf{w}_i$ and by using $(1 - \eta\sigma_i^2) \le 1$, from $\eta \le 1/\sigma_{\max}^2$. This concludes the proof of equation (2).

## H  THE EXPECTED JACOBIAN FOR CONVOLUTIONAL GENERATORS

We first prove the closed form expression of the expected Jacobian, or more precisely of the matrix $\mathbf{\Sigma}(\mathbf{U})$ defined in (13). By the expression for the Jacobian in equation (9), we have that

$$\mathcal{J}(\mathbf{C})\mathcal{J}^T(\mathbf{C}) = \sum_{\ell=1}^k v_\ell^2 \mathrm{diag}(\sigma'(\mathbf{U}\mathbf{c}_\ell))\mathbf{U}\mathbf{U}^T \mathrm{diag}(\sigma'(\mathbf{U}\mathbf{c}_\ell))$$

$$= \sum_{\ell=1}^k v_\ell^2 \sigma'(\mathbf{U}\mathbf{c}_\ell)\sigma'(\mathbf{U}\mathbf{c}_\ell)^T \odot \mathbf{U}\mathbf{U}^T$$

$$= \sigma'(\mathbf{U}\mathbf{C})\mathrm{diag}(v_1^2, \ldots, v_k^2)\sigma'(\mathbf{U}\mathbf{C})^T \odot \mathbf{U}\mathbf{U}^T,$$

where $\odot$ denotes the entrywise product of the two matrices. Then,

$$\mathbb{E}\left[\mathcal{J}(\mathbf{C})\mathcal{J}^T(\mathbf{C})\right] = \sum_{\ell=1}^k v_\ell^2 \mathbb{E}\left[\sigma'(\mathbf{U}\mathbf{c}_\ell)\sigma'(\mathbf{U}\mathbf{c}_\ell)^T\right] \odot \mathbf{U}\mathbf{U}^T. \tag{40}$$

Next, we have with (Daniely et al., 2016, Sec. 4.2) and using that the derivative of the ReLU function is the step function,

$$\left[\mathbb{E}\left[\sigma'(\mathbf{U}\mathbf{c}_\ell)\sigma'(\mathbf{U}\mathbf{c}_\ell)^T\right]\right]_{ij} = \frac{1}{2}\left(1 - \cos^{-1}\left(\frac{\langle \mathbf{u}_i, \mathbf{u}_j \rangle}{\|\mathbf{u}_i\|_2\|\mathbf{u}_j\|_2}\right)/\pi\right).$$

Using that $\|\mathbf{v}\|_2 = 1$, we get

$$\left[\mathbb{E}\left[\mathcal{J}(\mathbf{C})\mathcal{J}^T(\mathbf{C})\right]\right]_{ij} = \frac{1}{2}\left(1 - \cos^{-1}\left(\frac{\langle \mathbf{u}_i, \mathbf{u}_j \rangle}{\|\mathbf{u}_i\|_2\|\mathbf{u}_j\|_2}\right)/\pi\right)\langle \mathbf{u}_i, \mathbf{u}_j \rangle,$$

where $\mathbf{u}_i$ are the rows of $\mathbf{U}$. This concludes the proof of equation (13).

We next briefly comment on the singular value decomposition of a circulant matrix and explain that the singular vectors are given by Definition 1. Recall, that $\mathbf{U} \in \mathbb{R}^{n \times n}$ is a circulant matrix, implementing the convolution with a filter. Assume for simplicity that $n$ is even. It is well known that the discrete Fourier transform diagonalizes $\mathbf{U}$, i.e.,

$$\mathbf{U} = \mathbf{F}^{-1}\hat{\mathbf{U}}\mathbf{F},$$

where $\mathbf{F} \in \mathbb{C}^{n \times n}$ is the DFT matrix with entries

$$[\mathbf{F}]_{jk} = e^{i2\pi jk/n}, \quad j, k = 0, \ldots, n-1,$$

and $\hat{\mathbf{U}}$ is a diagonal matrix with diagonal $\mathbf{F}\mathbf{u}$, where $\mathbf{u}$ is the first column of the circulant matrix $\mathbf{U}$. From this, we can compute the singular value decomposition of $\mathbf{U}$ by using that $\hat{\mathbf{u}} = \mathbf{F}\mathbf{u}$ is conjugate symmetric (since $\mathbf{u}$ is real) so that

$$[\hat{\mathbf{u}}]_{n-k+2} = \hat{\mathbf{u}}^*, \quad k = 2, \ldots, n/2.$$

Let $\mathbf{U} = \mathbf{W}\mathbf{\Sigma}\mathbf{V}^T$ be the singular value decomposition of $\mathbf{U}$. The entries of the left singular vectors are given by the trigonometric basis functions defined in (5), and the singular values are given by the absolute values of $\hat{\mathbf{u}}$.

# I  PROOFS OF LEMMAS FOR NEURAL NETWORK DENOISERS (PROOFS OF AUXILIARY LEMMAS IN SECTION F)

## I.1  PROOF OF LEMMA 7: JACOBIAN PERTURBATION AROUND INITIALIZATION

The proof follows that of (Oymak & Soltanolkotabi, 2019b, Lem. 6.9).

**Step 1:**  We start by relating the perturbation of the Jacobian to a perturbation of the activation patterns. For any $\mathbf{C}, \mathbf{C}'$, we have that

$$\|\mathcal{J}(\mathbf{C}) - \mathcal{J}(\mathbf{C}')\| \leq \|\mathbf{v}\|_\infty \|\mathbf{U}\| \max_j \|\sigma'(\mathbf{u}_j^T \mathbf{C}) - \sigma'(\mathbf{u}_j^T \mathbf{C}')\|^2. \tag{41}$$

To see this, first note that, by (9),

$$\mathcal{J}(\mathbf{C}) - \mathcal{J}(\mathbf{C}') = [\ldots v_j(\mathrm{diag}(\sigma'(\mathbf{U}\mathbf{c}_j)) - \mathrm{diag}(\sigma'(\mathbf{U}\mathbf{c}_j))')\mathbf{U} \ldots].$$

This in turn implies that

$$\begin{aligned}
\|\mathcal{J}(\mathbf{C}) - \mathcal{J}(\mathbf{C}')\|^2 &= \left\|(\mathcal{J}(\mathbf{C}) - \mathcal{J}(\mathbf{C}'))(\mathcal{J}(\mathbf{C}) - \mathcal{J}(\mathbf{C}'))^T\right\| \\
&= \left\|(\sigma'(\mathbf{U}\mathbf{C}) - \sigma'(\mathbf{U}\mathbf{C}'))\mathrm{diag}(v_1^2, \ldots, v_k^2)(\sigma'(\mathbf{U}\mathbf{C}) - \sigma'(\mathbf{U}\mathbf{C}'))^T \odot \mathbf{U}\mathbf{U}^T\right\|^2 \\
&\overset{(i)}{\leq} \|\mathbf{U}\|^2 \max_j \left\|(\sigma'(\mathbf{u}_j^T \mathbf{C}) - \sigma'(\mathbf{u}_j^T \mathbf{C}'))\mathrm{diag}(\mathbf{v})\right\|^2 \\
&\leq \|\mathbf{v}\|_\infty^2 \|\mathbf{U}\|^2 \max_j \left\|\sigma'(\mathbf{u}_j^T \mathbf{C}) - \sigma'(\mathbf{u}_j^T \mathbf{C}')\right\|^2,
\end{aligned}$$

where for (i) we used that for two positive semidefinite matrices $\mathbf{A}, \mathbf{B}$, $\lambda_{\max}(\mathbf{A} \odot \mathbf{B}) \leq \lambda_{\max}(\mathbf{A}) \max_i \mathbf{B}_{ii}$. This concludes the proof of equation (41).

**Step 2:**  Step one implies that we only need to control $\sigma'(\mathbf{U}\mathbf{c}_j)$ around a neighborhood of $\mathbf{c}_j'$. Since $\sigma'$ is the step function, we need to count the number of sign flips between the matrices $\mathbf{U}\mathbf{C}$ and $\mathbf{U}\mathbf{C}'$. Let $|\mathbf{v}|_{\pi(q)}$ be the $q$-th smallest entry of $\mathbf{v}$ in absolute value.

**Lemma 8.** *Suppose that, for all $i$, and $q \leq k$,*

$$\|\mathbf{C} - \mathbf{C}'\| \leq \sqrt{q}\frac{|\mathbf{u}_i^T \mathbf{C}'|_{\pi(q)}}{\|\mathbf{u}_i\|}.$$

*Then*

$$\max_i \left\|\sigma'(\mathbf{u}_i^T \mathbf{C}) - \sigma'(\mathbf{u}_i^T \mathbf{C}')\right\| \leq \sqrt{2q}$$

*Proof.* Suppose that $\sigma'(\mathbf{u}_i^T \mathbf{C})$ and $\sigma'(\mathbf{u}_i^T \mathbf{C}')$ have $2q$ many different entries, then the conclusion of the statement would be violated. We show that this implies the assumption is violated as well, proving the statement by contraction. By the contradiction hypothesis,

$$\begin{aligned}
\|\mathbf{C} - \mathbf{C}'\|^2 &\geq \left\|\mathbf{u}_i^T (\mathbf{C} - \mathbf{C}')\right\|^2 / \|\mathbf{u}_i\|^2 \\
&\geq q\frac{|\mathbf{u}_i^T \mathbf{C}'|_{\pi(q)}^2}{\|\mathbf{u}_i\|^2},
\end{aligned}$$

where the last inequality follows by noting that at least $2q$ many entries have different signs, thus their difference is larger than their individual magnitudes, and at least $q$ many individual magnitudes are lower bounded by the $q$-th smallest one. $\qquad\square$

**Step 3:**  Next, we note that, with probability at least $1 - ne^{-kq^2/2}$, the $q$-th smallest entry of $\mathbf{u}_i^T \mathbf{C}' \in \mathbb{R}^k$ obeys

$$\frac{|\mathbf{u}_i^T \mathbf{C}'|_{\pi(q)}}{\|\mathbf{u}_i\|} \geq \frac{q}{2k}\nu \quad \text{for all } i = 1, \ldots, n. \tag{42}$$

We note that it is sufficient to prove this result for $\nu = 1$. This follows from anti-concentration of Gaussian random variables. Specifically, with the entries of $\mathbf{C}'$ being iid $\mathcal{N}(0,1)$ distributed, the entries of $\mathbf{g} = \mathbf{u}_i^T \mathbf{C}'/\|\mathbf{u}_i\| \in \mathbb{R}^k$ are iid standard Gaussian random variables as well. We show that with probability at least $1 - e^{-kq^2/2}$, at most $q$ entries are larger than $\frac{q}{2k}$. Let $\gamma_\delta$ be the number for which $\mathrm{P}\left[|g_\ell| \leq \gamma_\delta\right] \leq \delta$, where $g$ is a standard Gaussian random variable. Note that $\gamma_\delta \geq \sqrt{\pi/2}\delta \geq \delta$. Define the random variable

$$\delta_\ell = \begin{cases} 1 & \text{if } |g_\ell| \leq \gamma_\delta, \\ 0 & \text{otherwise.} \end{cases}$$

with $\delta = \frac{q}{2k}$. With $\mathbb{E}\left[\delta_\ell\right] = \delta$, by Hoeffding's inequality,

$$\mathrm{P}\left[\sum_{\ell=1}^{k} \delta_\ell \geq m\right] = \mathrm{P}\left[\sum_{\ell=1}^{k} \delta_\ell - \mathbb{E}\left[\delta_\ell\right] \geq m/2\right] \leq e^{-2k(m/2)^2} = e^{-km^2/2}. \tag{43}$$

Thus, with probability at least $1 - ke^{-kq^2/2}$ no more than $m$ entries are smaller than $\gamma_\delta \geq \delta = \frac{q}{2k}$. The results now follows by taking the union bound over all $i = 1, \ldots, n$.

We are now ready to conclude the proof of the lemma. By equation (41),

$$\|\mathcal{J}(\mathbf{C}) - \mathcal{J}(\mathbf{C}')\| \leq \|\mathbf{v}\|_\infty \|\mathbf{U}\| \max_j \left\|\sigma'(\mathbf{u}_j^T \mathbf{C}) - \sigma'(\mathbf{u}_j^T \mathbf{C}')\right\|$$

$$\leq \|\mathbf{v}\|_\infty \|\mathbf{U}\| \sqrt{2q}$$

provided that

$$\|\mathbf{C} - \mathbf{C}'\| \leq \sqrt{q}\frac{q}{2k}\nu,$$

with probability at least $1 - ne^{-kq^2/2}$. Setting $q = (2kR)^{2/3}$ concludes the proof (note that the assumption $R \leq \frac{1}{2}\sqrt{k}$ ensures $q \leq k$).

## I.2 PROOF OF LEMMA 5: BOUNDED JACOBIAN

By the expression of the Jacobian in equation (9),

$$\|\mathcal{J}(\mathbf{C})\|^2 = \left\|\mathcal{J}(\mathbf{C})\mathcal{J}(\mathbf{C})^T\right\|$$

$$= \left\|\sigma'(\mathbf{U}\mathbf{C})\mathrm{diag}(v_1^2, \ldots, v_k^2)\sigma'(\mathbf{U}\mathbf{C})^T \odot \mathbf{U}\mathbf{U}^T\right\|^2$$

$$\overset{(i)}{\leq} \|\mathbf{U}\|^2 \max_j \left\|\sigma'(\mathbf{u}_j^T \mathbf{C})\mathrm{diag}(\mathbf{v})\right\|_2^2$$

$$\leq \|\mathbf{v}\|_2^2 \|\mathbf{U}\|^2,$$

where for (i) we used that for two positive semidefinite matrices $\mathbf{A}, \mathbf{B}$, $\lambda_{\max}(\mathbf{A} \odot \mathbf{B}) \leq \lambda_{\max}(\mathbf{A}) \max_i \mathbf{B}_{ii}$. To prove the second inequality note that

$$\left\|\mathbf{J}\mathbf{J}^T\right\| = \|\mathbf{\Sigma}(\mathbf{U})\|$$

$$= \|\mathbf{v}\|_2^2 \left\|\mathbb{E}_{\mathbf{c}\sim\mathcal{N}(0,\mathbf{I})}\left[\mathrm{ReLU}'(\mathbf{U}\mathbf{c})\mathrm{ReLU}'(\mathbf{U}\mathbf{c})^T\right] \odot \left(\mathbf{U}\mathbf{U}^T\right)\right\|$$

$$\overset{(i)}{\leq} \|\mathbf{v}\|_2^2 \left\|\mathbf{U}\mathbf{U}^T\right\|.$$

Here, (i) follows from the fact that for two positive semidefinite matrices $\mathbf{A}, \mathbf{B}$, $\lambda_{\max}(\mathbf{A} \odot \mathbf{B}) \leq \lambda_{\max}(\mathbf{A}) \max_i \mathbf{B}_{ii}$.

## I.3 PROOF OF LEMMA 3: CONCENTRATION LEMMA

We begin by defining the zero-mean random matrices

$$\mathbf{S}_\ell = v_\ell^2 \left(\sigma'(\widetilde{\mathbf{U}}\mathbf{c}_\ell)\sigma'(\widetilde{\mathbf{U}}\mathbf{c}_\ell)^T - \mathbb{E}\left[\sigma'(\widetilde{\mathbf{U}}\mathbf{c}_\ell)\sigma'(\widetilde{\mathbf{U}}\mathbf{c}_\ell)^T\right]\right) \odot \left(\widetilde{\mathbf{U}}\widetilde{\mathbf{U}}^T\right).$$

With this notation,

$$
\mathcal{J}(\mathbf{C})\mathcal{J}^T(\mathbf{C}) - \mathbf{\Sigma}(\widetilde{\mathbf{U}}) = \sum_{\ell=1}^{k} v_\ell^2 \left( \sigma'(\widetilde{\mathbf{U}}\mathbf{c}_\ell)\sigma'(\widetilde{\mathbf{U}}\mathbf{c}_\ell)^T - \mathbb{E}\left[ \sigma'(\widetilde{\mathbf{U}}\mathbf{c}_\ell)\sigma'(\widetilde{\mathbf{U}}\mathbf{c}_\ell)^T \right] \right) \odot \left( \widetilde{\mathbf{U}}\widetilde{\mathbf{U}}^T \right)
$$

$$
= \sum_{\ell=1}^{k} \mathbf{S}_\ell.
$$

To show concentration we use the matrix Hoeffding inequality. To this aim note that the summands are centered in the sense that $\mathbb{E}\left[\mathbf{S}_\ell\right] = \mathbf{0}$. Next note that

$$
\left( \sigma'(\widetilde{\mathbf{U}}\mathbf{c}_\ell)\sigma'(\widetilde{\mathbf{U}}\mathbf{c}_\ell)^T - \mathbb{E}\left[ \sigma'(\widetilde{\mathbf{U}}\mathbf{c}_\ell)\sigma'(\widetilde{\mathbf{U}}\mathbf{c}_\ell)^T \right] \right) \odot \left( \widetilde{\mathbf{U}}\widetilde{\mathbf{U}}^T \right) \preceq \left( \sigma'(\widetilde{\mathbf{U}}\mathbf{c}_\ell)\sigma'(\widetilde{\mathbf{U}}\mathbf{c}_\ell)^T \right) \odot \left( \widetilde{\mathbf{U}}\widetilde{\mathbf{U}}^T \right)
$$

$$
= \mathrm{diag}\left( \sigma'(\widetilde{\mathbf{U}}\mathbf{c}_\ell)^T \right) \widetilde{\mathbf{U}}\widetilde{\mathbf{U}}^T \mathrm{diag}\left( \sigma'(\widetilde{\mathbf{U}}\mathbf{c}_\ell) \right)
$$

$$
\preceq B^2 \widetilde{\mathbf{U}}\widetilde{\mathbf{U}}^T
$$

Similarly,

$$
\left( \sigma'(\widetilde{\mathbf{U}}\mathbf{c}_\ell)\sigma'(\widetilde{\mathbf{U}}\mathbf{c}_\ell)^T - \mathbb{E}\left[ \sigma'(\widetilde{\mathbf{U}}\mathbf{c}_\ell)\sigma'(\widetilde{\mathbf{U}}\mathbf{c}_\ell)^T \right] \right) \odot \left( \widetilde{\mathbf{U}}\widetilde{\mathbf{U}}^T \right) \succeq - \mathbb{E}\left[ \sigma'(\widetilde{\mathbf{U}}\mathbf{c}_\ell)\sigma'(\widetilde{\mathbf{U}}\mathbf{c}_\ell)^T \right] \odot \left( \widetilde{\mathbf{U}}\widetilde{\mathbf{U}}^T \right)
$$

$$
\succeq - B^2 \widetilde{\mathbf{U}}\widetilde{\mathbf{U}}^T.
$$

Thus,

$$
-v_\ell^2 B^2 \widetilde{\mathbf{U}}\widetilde{\mathbf{U}}^T \preceq \mathbf{S}_\ell \preceq v_\ell^2 B^2 \widetilde{\mathbf{U}}\widetilde{\mathbf{U}}^T.
$$

Therefore, using $\mathbf{A}_\ell := v_\ell^2 B^2 \widetilde{\mathbf{U}}\widetilde{\mathbf{U}}^T$ we have

$$
\mathbf{S}_\ell^2 \preceq \mathbf{A}_\ell^2,
$$

and

$$
\sigma^2 := \left\| \sum_{\ell=1}^{k} \mathbf{A}_\ell^2 \right\| \leq B^4 \left( \sum_{\ell=1}^{k} v_\ell^4 \right) \left\| \widetilde{\mathbf{U}} \right\|^4
$$

To continue we will apply matrix Hoeffding inequality stated below.

**Theorem 6** (Matrix Hoeffding inequality, Theorem 1.3 (Tropp, 2011)). *Consider a finite sequence* $\mathbf{S}_\ell$ *of independent, random, self-adjoint matrices with dimension* $n$, *and let* $\{\mathbf{A}_\ell\}$ *be a sequence of fixed self-adjoint matrices. Assume that each random matrix satisfies*

$$
\mathbb{E}[\mathbf{S}_\ell] = \mathbf{0} \quad and \quad \mathbf{S}_\ell^2 \preceq \mathbf{A}_\ell^2 \quad almost \ surely.
$$

*Then, for all* $t \geq 0$,

$$
\mathrm{P}\left[ \left\| \sum_{\ell=1}^{k} \mathbf{S}_\ell \right\| \geq t \right] \leq 2n e^{-\frac{t^2}{8\sigma^2}} \quad where \quad \sigma^2 := \left\| \sum_{\ell=1}^{k} \mathbf{A}_\ell^2 \right\|.
$$

Therefore, applying matrix Hoeffding inequality we get that

$$
\mathrm{P}\left[ \left\| \sum_{\ell=1}^{k} \mathbf{S}_\ell \right\| \geq t \right] \leq 2n e^{-\frac{t^2}{B^4 \|\mathbf{v}\|_4^4 \|\widetilde{\mathbf{U}}\|^4}},
$$

which concludes the proof.

### I.4 PROOF OF LEMMA 6 (BOUND ON INITIAL RESIDUAL)

Without loss of generality we prove the result for $\nu = 1$. First note that by the triangle inequality

$$
\|\mathbf{r}_0\|_2 = \|\sigma(\mathbf{U}\mathbf{C})\mathbf{v} - \mathbf{y}\|_2 \leq \|\sigma(\mathbf{U}\mathbf{C})\mathbf{v}\|_2 + \|\mathbf{y}\|_2.
$$

We next bound $\|\sigma(\mathbf{UC})\mathbf{v}\|_2$. Consider the $i$-th entry of the vector $\sigma(\mathbf{UC})\mathbf{v} \in \mathbb{R}^n$, given by $\sigma(\mathbf{u}_i^T\mathbf{C})\mathbf{v}$, and note that $q_j = (\sigma(\mathbf{u}_i^T\mathbf{c}_j) - \sigma(\mathbf{u}_i^T\mathbf{c}_{n-j}))/\|\mathbf{u}_i\|_2$ is sub-Gaussian with parameter 2, i.e., $\mathrm{P}\left[|q_j| \geq t\right] \leq 2e^{-2t^2}$. It follows that

$$\mathrm{P}\left[\left|\sum_{j=1}^{k/2} q_j\right| \geq \beta\sqrt{k}\right] \leq 2e^{-\frac{\beta^2}{8}}.$$

Thus,

$$\mathrm{P}\left[\left|\sigma(\mathbf{u}_i^T\mathbf{C})\mathbf{v}\right| \geq \|\mathbf{u}_i\|_2\xi\beta\right] \leq 2e^{-\frac{\beta^2}{8}},$$

where we used that $|v_j| = \xi/\sqrt{k}$. Taking a union bound over all $n$ entries,

$$\mathrm{P}\left[\|\sigma(\mathbf{UC})\mathbf{v}\|_2^2 \geq \|\mathbf{U}\|_F^2\,\xi^2\beta^2\right] \leq 2ne^{-\frac{\beta^2}{8}}.$$

Choosing $\beta = \sqrt{8\log(2n/\delta)}$ concludes the proof.

