# OpenReview forum: "Denoising and Regularization via Exploiting the Structural Bias of Convolutional Generators"
_ICLR.cc/2020/Conference — Accept (Poster)_

### Official Review · AnonReviewer3 · 2019-10-16
**Official Blind Review #3**

**Rating:** 6

**Review:**

The paper provides a theoretical study of regularization capabilities of over-parameterized convolutional generators trained via gradient descent, in the context of denoising with an approach similar to the "deep image prior".
The authors show that when using appropriate upsampling operators, gradient descent biases the reconstructed images towards low-frequency components, while the noise components, which typically consist of high-frequency patterns, take longer to fit, so that early stopping can provide a useful bias for denoising.
The proofs are based on recent results for "lazy" training of over-parameterized networks, namely neural tangent kernels.

I find the paper interesting and novel, as it is the first to my knowledge to study the implicit bias of optimization for such generator-based denoising approaches, and provides interesting theoretical and empirical denoising results.
While the current results are for a simple architecture with one hidden layer, they may pave the way for a formal study of more complex architectures.
Nevertheless, the paper presents some limitations which should be addressed further in the paper:

* in terms of denoising capabilities, it seems that the results for the non-linear, over-parameterized network are essentially similar to the linear case (with an appropriately designed operator J in Section 3). Are there any benefits to the non-linear case that I am missing? Whether or not this is a limitation of the study, it should be discussed further in the paper.

* Some comments on the empirical results reported in Figure 2: (i) how are hyper-parameters chosen? this seems to be crucial for the effective use of such denoising strategies, and it is unclear how robust these methods are, e.g., to the early stopping time; (ii) there are other methods that only learn on the given image (thus without the need for a training set) which outperform BM3D [e.g. 1,2], it is unclear how the proposed methods compare to those.

other minor comments:
- Figure 2 "maintained on a large test set of images" more details are welcome, including on hyperparameter selection
- Section 1.1:
	* "demystifying" -> understanding?
	* different notations are used for the (same?) iterates, C^t and C_tau
- Section 2.3
	* "it follow that optimizing...": please clarify. I agree that the optimum is the same, but is the implicit bias from initialization the same?
- end of Section 3: early stoped -> stopped
- Section 4
	* after Definition 2, "this in turn implies that the jacobian spectrum ...rapidly": clarify, is it a consequence of the analysis?
	* after Thm 1: maybe give some insight about the proof and the use of NTK? also, discuss similarities or differences with the linear case


[1] Dabov et al. (2009) BM3D Image Denoising with Shape-Adaptive Principal Component Analysis
[2] Mairal et al. (2009) Non-local Sparse Models for Image Restoration

**Experience Assessment:**

I have published one or two papers in this area.

**Review Assessment: Checking Correctness Of Derivations And Theory:**

I assessed the sensibility of the derivations and theory.

**Review Assessment: Checking Correctness Of Experiments:**

I assessed the sensibility of the experiments.

**Review Assessment: Thoroughness In Paper Reading:**

I read the paper thoroughly.

---

> ### Author Response · Authors · 2019-11-12
> **Response to review 3**
>
> Thanks for reviewing our paper and your comments.
>
> Linear vs non-linear case: Yes, the results rely on association with a linear model. We have rewritten section 4.1 entitled ``Multilayer networks and moderate over-parameterization'' to point out limitations of this approach. Our results do not show a benefit of non-linear methods over linear ones. Specifically, our results explain why fitting a convolutional generator enables denoising a low-frequency signal, but does not explain why it denoises better in practice than a linear methods (such as thresholding). That said, we believe that our approach serves as a starting point to understand the more nonlinear behavior by tracking how the Jacobian matrix changes across time. That is, we hope to analyze the nonlinear case by successive linearization at different iterations in the future.
>
> Empirical results: We agree that the exact parameters of the empirical results are important and have therefore made available all our code to reproduce the figures here:
> https://www.dropbox.com/s/vtvavzry9sp5wrj/overparameterized_convolutional_generators-master.zip?dl=0
>
> With respect to comparisons with BM3D we have added a new appendix section with more quantitative experiments showing that neural networks based techniques can work better than BM3D. We were not aware of this particular BM3D variant mentioned by the reviewer, thanks for pointing this out. We were not able to compare with the particular variant of BM3D mentioned by the reviewer thus far as we could not find the code online and hence were not able to compare in the response period until now. We are happy to make such comparisons in the final version of the manuscript. That said, we would like to emphasize that in this paper we are not trying to say neural networks give the best performance for denoising (although the deep image prior paper and numerous other publications suggest this). Our goal in this paper is to begin to understant why they work so well and understand their behavior and limitations for denoising even if they might not be the best performing approach for a specific image or signal.
>
> Minor comments:
> Figure 1: ``[denoising performance, ordering of algorithms] maintained on a large test set of images'': We have added a section in the appendix (appendix A) explaining the experiment, and providing the number. Also, we have made the code available which enables reproducing the experiments.
> Section 1.1: We fixed the typos, thanks for pointing them out.
> Section 2.3: We agree that while the optimum is the same, the dynamics of gradient descent might be slightly different. However, our numerical results show that the phenomena that lower frequency components are fitted faster than high frequency ones can be reliably observed in both of those two extremely similar models.
> Section 4: ``This in turn implies that the Jacobian spectrum throughout training decreases rapidly'': We removed the sentence as the `throughout training' was misleading. What is correct is that the Jacobian spectrum at initialization decays rapidly.
> Section 4: As suggested by the reviewer, we have added a paragraph at the end of section 4 explaining the proof.

---

### Official Review · AnonReviewer2 · 2019-10-23
**Official Blind Review #2**

**Rating:** 8

**Review:**

This paper studies the theoretical reasons why a randomly initialized decoder or autoencoder like architecture can prove useful for image denoising, by using early stopping. The paper is clearly written and the proofs are interesting. It will be of interest to the community.

Some questions that are not addressed, and I'd be keen to understand are:

1. What is the effect of a bias in a layer?

2. What family of noise types (in this work only additive Gaussian noise was considered) that will benefit from early stopping? What about multiplicative noise ("shot noise"), for example (which often arises in computational photography)?

3. Unless I missed it, it was not clear to me that the analysis shed light on why a trained network performs much worse than a random network?

Notes:

1. Figure 1 could be improved by adding x-axis to both rows, and showing the exact difference in number of iterations between fixed learned filters. It seems that the difference happens in two ways: the natural images converge faster for fixed filters, *and* the noisy images converge slower; so that the gap between noisy and natural images is larger for fixed filters.

2. A number of typos, a missing equation reference etc. Please proofread.

**Experience Assessment:**

I have published in this field for several years.

**Review Assessment: Checking Correctness Of Derivations And Theory:**

I carefully checked the derivations and theory.

**Review Assessment: Checking Correctness Of Experiments:**

I carefully checked the experiments.

**Review Assessment: Thoroughness In Paper Reading:**

I read the paper thoroughly.

---

> ### Author Response · Authors · 2019-11-12
> **Response to review 2**
>
> Thanks for reviewing our paper and the comments.
>
> Question 1, regarding the effect of bias: In the model that we study theoretically, the output is given by relu(U C') v'. A corresponding model with bias would be relu(U C + B) v, where B is a matrix where each column is equal to a constant. In this model with biases each channel, i.e., each column of U C + B, would have its own independent bias. Both models are equivalent. To see this, note that because U has full rank, we can use every other channel to add a constant, i.e., there is a choice of coefficients c_i for a given channel i so that [b_i, b_i, …., b_i] = U c_i. A similar argument can be made for the deep decoder with multiple layers.
>
> Question 2, family of noise types: Our results as stated pertain to additive noise, but do not assume a particular distribution. Specifically, Theorem 2 shows that any component of the signal that is high-frequency is fitted later than the low-frequency components and thus any high-frequency component will be filtered out. So if a multiplicative noise adds high frequencies (such as speckle noise), then such noise components are filtered out as well.
>
> Question 3: Our analysis unfortunately does not shed light on why a trained network performs worse than  a random one. Two comments on that:
> i) Suppose we train a network end to end for denoising. If the image to be denoised comes from the same distribution as the images the network has been trained on, it can perform better than our method that assumes no prior knowledge on the images.
> ii) Our results rely on the network being randomly initialized. We have carried out simulations where we first optimize on one image and then plug in another, and that can hurt performance slightly. So initializing randomly is a good choice, both in theory and practice.
>
> Notes: Thanks for the notes, we proofread the manuscript and fixed typos and a broken reference. Thanks for pointing this out!

---

### Official Review · AnonReviewer1 · 2019-10-27
**Official Blind Review #1**

**Rating:** 6

**Review:**

This paper studies the situation in which a two-layer CNN with RELU nonlinearity is fit to a single image and the observation that it is able to fit a "natural" image in fewer iterations than a "noisy" image. Theorems on the convergence of this fitting are discussed and proven in the appendices. Intermediate results study the convergence of fitting a linear model to an image plus noise and fitting a single-layer CNN. Denoising is demonstrated on two images with additive white Gaussian noise and the approach under study is shown to provide a better signal-to-noise ratio than BM3D, another untrained denoising approach. The main result is that the use of upsampling via a fixed interpolation filter provides an inductive bias towards "natural" images.

The bibliography does a good job of positioning this paper within the recent set of articles exploring the curious behavior of fitting a CNN to a single image. The problem is interesting, timely, and surprising. The analysis does provide some insight into what is going on.

But, the paper would do well to embrace the Fourier domain and first discuss what is meant by "natural images" and "noise" in terms of their frequency content. In particular, something like Simoncelli and Olshausen (2001) could be used as a description of the spectra of natural images. Additive white Gaussian noise has a flat spectrum, which is never mentioned in the paper. Thus, the theoretical result mainly highlights the fact that natural images have a low-pass spectrum, while white noise has a flat spectrum, and CNNs using interpolation also have a low-pass spectrum in some sense. This makes sense from a frequency perspective, because interpolation increases the sampling rate of a signal without changing its frequency content, i.e., it adds high frequencies with no energy.

The "trigonometric basis" used in the paper, which consists of sines and cosines at each frequency, could be more cleanly described as a basis of complex exponentials, i.e., the Fourier basis, which doesn't require partitioning the basis into a cosine half and a sine half. The discussion of triangular and Gaussian smoothing functions has been very well explored in the signal processing literature discussing windowing functions and their Fourier transforms, e.g., Harris (1978). I don't think Figure 3 showing some sinusoids is necessary.

The main body of the paper goes into the 10th page, but the appendices make up another 18 pages. The main body of the paper does not include any of the proofs of the provided theorems. This seems rather excessive.


Harris, F. J. (1978). On the use of windows for harmonic analysis with the discrete Fourier transform. Proceedings of the IEEE, 66(1), 51-83.

Simoncelli, E. P., & Olshausen, B. A. (2001). Natural image statistics and neural representation. Annual review of neuroscience, 24(1), 1193-1216.


Minor comments:

Several citations are in the wrong form (\citet instead of \citep) throughout the paper.

There is a link to a figure in the appendix that is broken on page 5

In figure 7, one one set of y-axis labels is shown, but it appears that each subplot uses an independent y-axis, just without labels. Please plot them all on the same y-axis.

Similarly, figures 1, 4, and 6 show similar things on different y-axes when they could be plotted at the same scale on the y-axis to make them more easily comparable visually.

Typos: "over-paramtrized" and "spacial" both on page 1.


After discussion:

The authors have addressed my concerns, so I am changing my decision from Weak Reject to Weak Accept.

**Experience Assessment:**

I have read many papers in this area.

**Review Assessment: Checking Correctness Of Derivations And Theory:**

I assessed the sensibility of the derivations and theory.

**Review Assessment: Checking Correctness Of Experiments:**

I assessed the sensibility of the experiments.

**Review Assessment: Thoroughness In Paper Reading:**

I read the paper at least twice and used my best judgement in assessing the paper.

---

> ### Author Response · Authors · 2019-11-12
> **Response to reviewer 1**
>
> Many thanks for the suggestion to point out the different Fourier spectra of natural images and noise. We have incorporated this by adding the following paragraph in the introduction and the main part:
> ``Note that a smooth signal is a good model for a natural image since natural images are well approximated by low-frequency components. Specifically, Figure 4 in (Simoncelli and Olshausen, 2001) shows that the power spectrum of a natural image (i.e., the energy distribution by frequency) decays rapidly from low frequencies to high frequencies. In contrast, Gaussian noise has a flat power spectrum.''
>
> On the comment that ``the theoretical result mainly highlights the fact that natural images have a low-pass spectrum, while white noise has a flat spectrum, and CNNs using interpolation also have a low-pass spectrum in some senses'': We agree that the main theoretical contribution is to show that CNNs with interpolating filters fit a natural image faster than noise by showing that it fits the lower-frequency components faster than high frequency ones. The novelty of our contribution is to i) rigorously prove that low-frequency components are fitted faster than high-frequency ones and ii) to attribute this effect to the convolution with interpolation filters.  Both contributions i and ii are new. That said, we emphasize that our results continue to explain denoising performance even if the noise does not have a completely flat spectrum as discussed on page 3. Our denoising result holds as long as the noise is not too low pass (but doesn’t have to be approximately flat).
>
> On the trigonometric basis functions: The trigonometric basis functions arise in our results because they are the left-singular vectors of a circulant matrix. As such, they are the unique choice for the left-singular vectors. Note that the Fourier basis diagonalizes a circulant matrix, but it is not equivalent to the left-singular vectors of a circulant matrix. Thus we cannot substitute the trigonometric basis functions with the Fourier basis vectors as suggested by the reviewer. Note that those basis functions are different objects in that one is a basis for real-valued vectors and the other for complex valued vectors.
>
> On triangular smoothing functions: We fully agree that triangular and Gaussian functions and their spectrum are discussed in the signal processing community. Note that our results do not depend on the Fourier transform of the kernels but on the associated weights/dual kernel defined in Definition 2.
>
> Minor comments: Many thanks for the comments, we have
> i) fixed the citations from cite to citep,
> ii) fixed the broken link to the figure,
> iii) changed Figure 7 so that each sub-plot is plotted on the same y-axis,
> iv) and fixed the two typos on page 1.

---

> > ### Comment · AnonReviewer1 · 2019-11-14
> > **Updates have improved the paper, but not addressed all of my concerns**
> >
> > I thank the authors for their updates to the paper, it has been improved.
> >
> > While I agree with them on the contribution of the paper, and I appreciate it novelty with respect to the analysis of untrained CNNs for image denoising, I still disagree on its rigor. I would find it much more rigorous to derive convergence rates as a function of signal-to-noise ratio at any particular spatial frequency as opposed to vague notions of "natural images" and noise that is "not too low pass (but doesn't have to be approximately flat)."

---

> > > ### Author Response · Authors · 2019-11-15
> > > **Addressing the final concern**
> > >
> > > Many thanks for the comment and for going through our revision. With this response we are trying to address the last concern.
> > > Our theorems provide rigours statements from which a convergence rate as a function of the signal-to-noise ratio at any particular frequency $i$, can be obtained, as asked for in the comment above.
> > > Specifically, a slight variation of Theorem 2 (obtained from equation (iv) in the paragraph in the proof entitled ``concluding the proof of Theorem 3'') yields that, under exactly the same conditions as Theorem 2, the estimate after $\tau$ iterations of gradient descent, given by $G(\mathbf C_\tau)$, obeys
> > > $$
> > > \left|\langle G(\mathbf C_\tau) - \mathbf x, \mathbf w_i \rangle \right|^2
> > > \leq
> > > \left|\langle \mathbf w_i,\mathbf x_i\rangle \right|^2 [  (1-\eta \sigma_i^2)^\tau  + \frac{1}{\sqrt{\mathrm{SNR}_i}} ( (1-\eta \sigma_i^2)^\tau  -1 )  ]^2
> > > + \epsilon.
> > > $$
> > > Here, $\mathrm{SNR}_i = \left|\langle \mathbf w_i,\mathbf x\rangle \right|^2 / \left|\langle \mathbf w_i,\mathbf z \rangle \right|^2$ is the signal-to-noise ratio of a particular spatial frequency. Note that the right hand side is the error of the prediction $G(\mathbf C_\tau)$ in the direction of the spatial frequency i, and the right hand side gives the convergence in terms of the SNR at the particular spatial frequency. We hope that this statement addresses the reviewer's concern.

---

### Decision · Program_Chairs · 2019-12-19

**Decision:**

Accept (Poster)

**Comment:**

This paper studies the question of why a network trained to reproduce a single image often de-noises the image early in training. This an interesting question and, post discussion, all three reviewers agree that it will be of general interest to the community and is worth publishing. Therefore I recommend it be accepted.